# Causal Feature Learning via Generalized Rayleigh Quotients

**Liang Cao**[1]  **Jun Wan**[2]  **Yan Qin**[3]  **Weide Liu**[2]

## Abstract

Extracting causally meaningful features from time-series data is fundamental for robust machine learning under distribution shifts. In process monitoring, existing methods struggle to maintain detection performance when operating conditions change. Current approaches capture either temporal causal relationships or cross-environment invariance, but not both simultaneously. We propose Causal Feature Learning (CFL), a unified framework that jointly optimizes for temporal relevance and environment mean invariance. CFL formulates feature extraction as a generalized Rayleigh-quotient problem, maximizing correlation with target variables while penalizing sensitivity to environment-dependent mean shifts. Theoretical analysis establishes conditions under which CFL identifies a mean-invariant predictive subspace. Experiments on the Tennessee Eastman Process demonstrate that CFL achieves 93.69% average fault detection rate, outperforming 15 baseline methods and validating the benefit of jointly capturing both aspects of causality.

## 1. Introduction

Discovering causal relationships from observational time-series data is a fundamental challenge in machine learning and process engineering (Pearl, 2009; Spirtes et al., 2000). In industrial process monitoring, understanding the causal structure among process variables enables robust fault detection that generalizes across different operating conditions (Chiang et al., 2000; Qin, 2012). However, most existing methods focus on statistical correlations rather than causal relationships, leading to features that may fail when

[1]Department of Chemical and Biological Engineering, University of British Columbia, Vancouver, BC, Canada [2]School of Computing and Artificial Intelligence, Jiangxi University of Finance and Economics, Nanchang, China [3]Chongqing University, Chongqing, China. Correspondence to: Weide Liu < weide001@e.ntu.edu.sg>.

*Proceedings of the 43rd International Conference on Machine Learning*, Seoul, South Korea. PMLR 306, 2026. Copyright 2026 by the author(s).

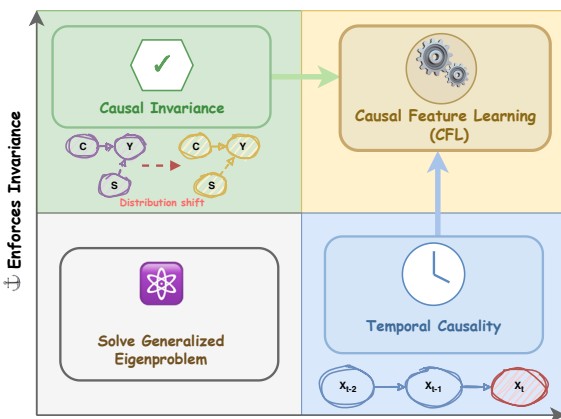

*Figure 1.* Two complementary aspects of causal features. (a) Temporal causality: time-lagged covariances encode how past variables help predict current states. (b) Causal invariance: causal mechanisms are expected to be stable across environments. CFL extracts features capturing both properties.

the underlying data distribution shifts.

Causal features in time-series data manifest through two complementary mechanisms, as illustrated in Figure 1. First, temporal causality reflects how past values of one variable influence current values of another, formalized through concepts like Granger causality (Granger, 1969). This temporal information is encoded in time-lagged covariance structures. If $X_{t-k}$ helps predict and potentially causes $Y_t$, their lagged cross-covariance can be informative. Second, causal invariance reflects the principle that causal mechanisms remain autonomous across environments or operating conditions (Peters et al., 2017; Schölkopf et al., 2012). In this work, we operationalize invariance via first-order invariance across environments: spurious correlations often manifest as environment-dependent mean shifts, while causal mechanisms are expected to be more stable.

Existing dynamic latent variable methods capture these aspects separately. Slow Feature Analysis (SFA) (Wiskott & Sejnowski, 2002) extracts features that vary slowly over time, implicitly assuming that meaningful signals are temporally smooth while noise fluctuates rapidly. Dynamic methods like DiPCA (Ku et al., 1995) and DiCCA (Dong & Qin,

2018) model temporal dependencies through autoregressive structures, capturing Granger-type predictive effects. However, neither approach explicitly optimizes for both temporal relevance and robustness to cross-environment mean shifts.

In this paper, we propose causal feature learning, a principled framework that jointly captures temporal relevance and mean invariance through generalized Rayleigh-quotient optimization. Our formulation balances two objectives: maximizing correlation with causally relevant target variables, and minimizing variation of feature means across operating environments. These objectives respectively capture temporal predictive effects and encourage invariance across operating conditions.

The contributions of this work are as follows:

- We identify two complementary aspects of causal features in time-series: temporal relevance encoded in lagged covariances and causal invariance reflected in cross-environment mean stability.

- We develop CFL, a generalized Rayleigh-quotient framework that jointly maximizes target relevance and penalizes environment-dependent mean shifts.

- We provide theoretical analysis establishing conditions under which CFL identifies a mean-invariant predictive subspace.

- We demonstrate on the Tennessee Eastman Process that CFL achieves state-of-the-art fault detection, outperforming 15 baseline methods.

## 2. Related Work

### 2.1. Temporal Causality and Granger Causality

Granger causality (Granger, 1969) formalizes temporal predictive relationships: $X$ Granger-causes $Y$ if past values of $X$ help predict $Y$ beyond what past values of $Y$ alone provide. This concept has been extended to multivariate settings (Lütkepohl, 2005) and nonlinear systems (Tank et al., 2022). Recent work on causal discovery from time series (Runge et al., 2019) has developed methods like PCMCI that combine constraint-based causal discovery with temporal structure.

In process monitoring, temporal effects are crucial because faults propagate through the process with characteristic delays determined by physical dynamics. Dynamic latent variable methods implicitly capture temporal predictability. DiPCA (Ku et al., 1995) constructs time-lagged feature matrices and maximizes covariance between current latent scores and predictions from past values:

$$\max_{\boldsymbol{w}} \mathrm{Cov}(\boldsymbol{w}^\top \boldsymbol{x}_t, \hat{\boldsymbol{w}}^\top \boldsymbol{x}_{t-1:t-s}). \qquad (1)$$

DiCCA (Dong & Qin, 2018) similarly maximizes correlation:

$$\max_{\boldsymbol{w}} \mathrm{Corr}(\boldsymbol{w}^\top \boldsymbol{x}_t, \hat{\boldsymbol{w}}^\top \boldsymbol{x}_{t-1:t-s}). \qquad (2)$$

DiPLS (Y.Dong & Qin, 2015) incorporates output variables, maximizing covariance between outputs and predicted latent scores. These methods capture how past states help predict current states but do not explicitly encourage invariance across operating conditions.

### 2.2. Causal Invariance and Independent Mechanisms

The principle of independent causal mechanisms (Peters et al., 2017; Schölkopf et al., 2012) motivates invariant causal prediction (ICP) (Peters et al., 2016) and invariant risk minimization (IRM) (Arjovsky et al., 2019), which identify causal features by finding representations whose predictive relationships are invariant across environments. Recent work has extended these ideas to heterogeneous and nonstationary data (Huang et al., 2020). In this paper, we focus on a tractable spectral formulation that enforces first-order (mean) invariance across environments.

Slow Feature Analysis (SFA) (Wiskott & Sejnowski, 2002) provides a complementary perspective. By extracting slowly varying features:

$$\min_{\boldsymbol{W}} \frac{(\boldsymbol{W}^\top \boldsymbol{\Omega}_d \boldsymbol{W})}{(\boldsymbol{W}^\top \boldsymbol{\Sigma}_d \boldsymbol{W})}, \qquad (3)$$

where $\boldsymbol{\Omega}_d$ measures temporal variation rate and $\boldsymbol{\Sigma}_d$ measures variance, SFA assumes that meaningful mechanisms vary slowly. While related in spirit, SFA enforces temporal smoothness rather than cross-environment invariance.

### 2.3. Process Monitoring and Fault Detection

Statistical process monitoring has a rich history (Chiang et al., 2000; Qin, 2012; Ge et al., 2013). PCA-based methods (Jackson, 1991; Nomikos & MacGregor, 1994) project data onto principal directions and monitor $T^2$ and SPE statistics. Extensions incorporate dynamics (Ku et al., 1995), quality variables (Y.Dong & Qin, 2015), and nonlinear relationships (Kruger & Xie, 2012). However, these methods primarily capture statistical correlations without explicitly modeling causal structure, limiting robustness under distribution shift.

### 2.4. Bridging Temporal Relevance and Invariance

Our work bridges the temporal and invariance perspectives. We argue that robust causal features should satisfy both properties: (1) encode temporally predictive structure via time-lagged representations; and (2) be stable across environments, operationalized here as mean invariance. Existing methods optimize for one property but not both, potentially missing features that are temporally informative but

environment-sensitive, or stable but not temporally informative.

# 3. Problem Formulation

## 3.1. Setting and Notation

Let $\boldsymbol{X} \in^{N \times m}$ denote time-series observations with $N$ samples and $m$ variables. We denote target variables (key process indicators) as $\boldsymbol{Y} \in^{N \times L}$.

To capture temporal structure, we construct time-lagged feature vectors:

$$\boldsymbol{z}_t = [\boldsymbol{x}_t^\top, \boldsymbol{x}_{t-1}^\top, \ldots, \boldsymbol{x}_{t-s}^\top]^\top \in^{m(s+1)}, \qquad (4)$$

where $s$ is the lag order. Let $d = m(s + 1)$ be the dimension of $\boldsymbol{z}_t$. We consider data from multiple operating environments $= \{e_1, \ldots, e_K\}$, representing different operating modes, seasonal conditions, or process configurations.

## 3.2. Time-Lagged Covariance Structure

We define the (population) time-lagged covariance matrix

$$\boldsymbol{\Sigma}_d \triangleq (\boldsymbol{z}_t) = \mathbb{E}[(\boldsymbol{z}_t - \mathbb{E}[\boldsymbol{z}_t])(\boldsymbol{z}_t - \mathbb{E}[\boldsymbol{z}_t])^\top], \qquad (5)$$

and its sample estimate

$$\hat{\boldsymbol{\Sigma}}_d = \frac{1}{N - s - 1} \sum_{t=s+1}^{N} (\boldsymbol{z}_t - \bar{\boldsymbol{z}})(\boldsymbol{z}_t - \bar{\boldsymbol{z}})^\top. \qquad (6)$$

The matrix $\boldsymbol{\Sigma}_d$ is block-structured:

$$\boldsymbol{\Sigma}_d = \begin{bmatrix} \mathrm{Cov}(\boldsymbol{x}_t, \boldsymbol{x}_t) & \cdots & \mathrm{Cov}(\boldsymbol{x}_t, \boldsymbol{x}_{t-s}) \\ \vdots & \ddots & \vdots \\ \mathrm{Cov}(\boldsymbol{x}_{t-s}, \boldsymbol{x}_t) & \cdots & \mathrm{Cov}(\boldsymbol{x}_{t-s}, \boldsymbol{x}_{t-s}) \end{bmatrix}. \qquad (7)$$

The off-diagonal blocks $\mathrm{Cov}(\boldsymbol{x}_t, \boldsymbol{x}_{t-k})$ encode lagged cross-dependencies. We also define the difference covariance matrix:

$$\boldsymbol{\Omega}_d \triangleq (\boldsymbol{z}_t - \boldsymbol{z}_{t-1}) = \mathbb{E}[(\Delta\boldsymbol{z}_t - \mathbb{E}[\Delta\boldsymbol{z}_t])(\Delta\boldsymbol{z}_t - \mathbb{E}[\Delta\boldsymbol{z}_t])^\top],$$
$$(8)$$

where $\Delta\boldsymbol{z}_t = \boldsymbol{z}_t - \boldsymbol{z}_{t-1}$, and its sample estimate

$$\hat{\boldsymbol{\Omega}}_d = \frac{1}{N - s - 2} \sum_{t=s+2}^{N} (\boldsymbol{z}_t - \boldsymbol{z}_{t-1})(\boldsymbol{z}_t - \boldsymbol{z}_{t-1})^\top. \qquad (9)$$

## 3.3. Two Aspects of Causal Features

**Definition 3.1** (Temporal Target Relevance). A feature direction $\boldsymbol{w}$ is temporally relevant for target $\boldsymbol{Y}$ if the time-lagged projection $\boldsymbol{w}^\top \boldsymbol{z}_t$ has non-zero covariance with $\boldsymbol{Y}$:

$$\mathrm{Cov}(\boldsymbol{w}^\top \boldsymbol{z}_t, \boldsymbol{Y}) \neq \boldsymbol{0}. \qquad (10)$$

This indicates that the combination of current and past values encoded in $\boldsymbol{w}^\top \boldsymbol{z}_t$ carries information about $\boldsymbol{Y}$.

**Definition 3.2** (Environment Mean Invariance). A feature direction $\boldsymbol{w}$ is (first-order) environment-invariant if its mean is invariant across environments:

$$\mathbb{E}^e[\boldsymbol{w}^\top \boldsymbol{z}] = \mathbb{E}^{e'}[\boldsymbol{w}^\top \boldsymbol{z}] \quad \forall e, e' \in . \qquad (11)$$

Equivalently, if $\boldsymbol{\mu}_z^{(k)} \triangleq \mathbb{E}[\boldsymbol{z} \mid e = e_k]$, then $\boldsymbol{w}^\top \boldsymbol{\mu}_z^{(k)}$ is constant over $k$.

CFL as defined in this paper enforces mean invariance via a between-environment mean-shift penalty. Stronger distributional invariance (e.g., $P^e(\boldsymbol{w}^\top \boldsymbol{z})$ identical across $e$) would require additional penalties such as higher-moment matching or kernel discrepancies. Our goal is to find a projection $\boldsymbol{W}_s$ that extracts features satisfying both properties: temporally relevant and environment mean-invariant.

# 4. Methodology

## 4.1. Causal Signal and Invariance Matrices

To capture temporal relevance to targets, define

$$\boldsymbol{A}_y = \sum_{i=1}^{L} \boldsymbol{\Sigma}_{z y_i} \boldsymbol{\Sigma}_{y_i y_i}^{-1} \boldsymbol{\Sigma}_{y_i z}, \qquad (12)$$

where $\boldsymbol{\Sigma}_{z y_i} = \mathbb{E}[(\boldsymbol{z} - \mathbb{E}[\boldsymbol{z}])(y_i - \mathbb{E}[y_i])] \in^d$ and $\boldsymbol{\Sigma}_{y_i y_i} = \mathrm{Var}(y_i) = \sigma_{y_i}^2$. The matrix $\boldsymbol{A}_y$ aggregates directions that are linearly predictive of the targets. To penalize deviations from mean invariance, define

$$\boldsymbol{A}_e = \sum_{k=1}^{K} \pi_k (\boldsymbol{\mu}_z^{(k)} - \bar{\boldsymbol{\mu}}_z)(\boldsymbol{\mu}_z^{(k)} - \bar{\boldsymbol{\mu}}_z)^\top, \qquad (13)$$

where $\boldsymbol{\mu}_z^{(k)} = \mathbb{E}[\boldsymbol{z} \mid e = e_k]$, $\bar{\boldsymbol{\mu}}_z = \sum_k \pi_k \boldsymbol{\mu}_z^{(k)}$, and $\pi_k$ is the proportion of samples from environment $e_k$. Large projection onto $\boldsymbol{A}_e$ indicates environment-dependent mean shifts, suggesting spurious sensitivity.

## 4.2. Generalized Rayleigh-Quotient Formulation

Let $\boldsymbol{B} \triangleq \boldsymbol{A}_e + \epsilon \boldsymbol{I} \succ 0$ with $\epsilon > 0$. We extract $d_s$ features by solving the constrained trace maximization

$$\max_{\boldsymbol{W}_s \in^{d \times d_s}} (\boldsymbol{W}_s^\top \boldsymbol{A}_y \boldsymbol{W}_s) \quad \text{s.t.} \quad \boldsymbol{W}_s^\top \boldsymbol{B} \boldsymbol{W}_s = \boldsymbol{I}_{d_s}. \qquad (14)$$

For $d_s = 1$, (14) reduces to the standard generalized Rayleigh quotient $\max_{\boldsymbol{w} \neq 0} \frac{\boldsymbol{w}^\top \boldsymbol{A}_y \boldsymbol{w}}{\boldsymbol{w}^\top \boldsymbol{B} \boldsymbol{w}}$.

The objective in (14) balances two complementary terms that reflect the dual nature of causal features. The signal term $(\boldsymbol{W}_s^\top \boldsymbol{A}_y \boldsymbol{W}_s)$ measures how well the extracted features capture information predictive of the target variables, leveraging the time-lagged representation to encode temporal dependencies. Meanwhile, the invariance constraint $\boldsymbol{W}_s^\top \boldsymbol{B} \boldsymbol{W}_s = \boldsymbol{I}$ penalizes directions that exhibit

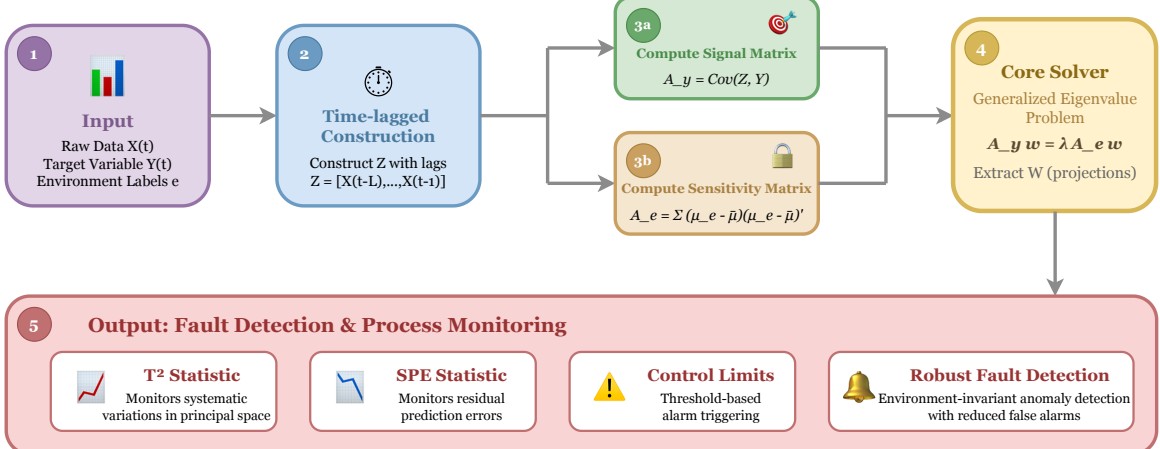

*Figure 2.* Framework of Causal Feature Learning via Generalized Rayleigh Quotients.

large environment-dependent mean shifts through the matrix $\boldsymbol{A}_e$, with the regularization parameter $\epsilon$ ensuring that $\boldsymbol{B}$ remains positive definite.

The KKT conditions yield the generalized eigenvalue problem

$$\boldsymbol{A}_y \boldsymbol{w} = \lambda \boldsymbol{B} \boldsymbol{w}. \tag{15}$$

Let $\{\boldsymbol{w}_i\}_{i=1}^{d_s}$ be the generalized eigenvectors associated with the largest eigenvalues $\lambda_1 \geq \cdots \geq \lambda_{d_s}$, normalized such that $\boldsymbol{W}_s^\top \boldsymbol{B} \boldsymbol{W}_s = \boldsymbol{I}_{d_s}$. Then $\boldsymbol{W}_s = [\boldsymbol{w}_1, \ldots, \boldsymbol{w}_{d_s}]$ solves (14).

### 4.3. Connection to Existing Methods

**Slow Feature Analysis.** SFA can be written in the standard constrained form

$$\min_{\boldsymbol{W}} (\boldsymbol{W}^\top \boldsymbol{\Omega}_d \boldsymbol{W}) \quad \text{s.t.} \quad \boldsymbol{W}^\top \boldsymbol{\Sigma}_d \boldsymbol{W} = \boldsymbol{I}. \tag{16}$$

Here, $\boldsymbol{\Omega}_d$ captures temporal variation via differences $\Delta \boldsymbol{z}_t$, and minimizing it extracts slowly varying features. The constraint $\boldsymbol{\Sigma}_d$ provides variance normalization that prevents degenerate solutions. While SFA uses temporal smoothness as a proxy for invariance, CFL explicitly penalizes cross-environment mean shifts via $\boldsymbol{A}_e$. This distinction matters when informative causal signals exhibit transients (e.g., during fault propagation) that SFA would suppress.

**Dynamic Latent Variable Methods.** DiPCA and DiCCA emphasize temporal predictability but do not explicitly control environment sensitivity; a direction may be strongly predictable yet driven by environment-dependent mean shifts. DiPLS incorporates target variables but maximizes predictive covariance without an explicit invariance penalty. In contrast, CFL balances target relevance (via $\boldsymbol{A}_y$) and mean invariance (via $\boldsymbol{A}_e$), yielding features that are both tem-porally informative and robust to environment-dependent mean shifts.

### 4.4. Fault Detection Based on CFL Features

Given the learned subspace $\text{span}(\boldsymbol{W}_s)$, we form an Euclidean-orthonormal basis

$$\boldsymbol{U}_s = (\boldsymbol{W}_s), \tag{17}$$

and compute feature scores $\boldsymbol{s}_t = \boldsymbol{U}_s^\top \boldsymbol{z}_t$.

$T^2$ **Statistic.** Let $\hat{\boldsymbol{\Sigma}}_s$ be the sample covariance of $\{\boldsymbol{s}_t\}$ on normal training data. We monitor

$$T_t^2 = \boldsymbol{s}_t^\top \hat{\boldsymbol{\Sigma}}_s^{-1} \boldsymbol{s}_t. \tag{18}$$

**SPE Statistic.** We monitor the residual energy orthogonal to the learned subspace:

$$\text{SPE}_t = \|(\boldsymbol{I} - \boldsymbol{U}_s \boldsymbol{U}_s^\top) \boldsymbol{z}_t\|^2. \tag{19}$$

The complete CFL framework is illustrated in Figure 2, and as described in Algorithm 1, CFL extracts features by solving a generalized eigenvalue problem.

## 5. Theoretical Analysis

### 5.1. Causal/Spurious Feature Decomposition

**Assumption 5.1** (Mean-Invariant Predictive vs. Spurious Mean-Shift Features)**.** There exists a direct-sum decomposition $^d = \mathcal{C} \oplus \mathcal{S}$ such that $\boldsymbol{z} = \boldsymbol{z}_c + \boldsymbol{z}_s$ with $\boldsymbol{z}_c \in \mathcal{C}$ and $\boldsymbol{z}_s \in \mathcal{S}$. The subspace $\mathcal{C}$ is mean-invariant in the sense that $\mathcal{C} \subseteq \ker(\boldsymbol{A}_e)$, i.e., $\boldsymbol{w}^\top(\boldsymbol{\mu}_z^{(k)} - \bar{\boldsymbol{\mu}}_z) = 0$ for all $\boldsymbol{w} \in \mathcal{C}$ and all $k$. The subspace $\mathcal{S}$ exhibits spurious mean shifts, i.e., there exist $e \neq e'$ such that $\mathbb{E}[\boldsymbol{z}_s \mid e] \neq \mathbb{E}[\boldsymbol{z}_s \mid e']$.

## Algorithm 1 Causal Feature Learning

**Require:** Time-series $\boldsymbol{X} \in^{N \times m}$, targets $\boldsymbol{Y} \in^{N \times L}$, environments $e \in \{e_1, \ldots, e_K\}$, lag order $s$, dimension $d_s$, regularization $\epsilon$
1: Construct time-lagged matrix $\boldsymbol{Z}$ (rows are $\boldsymbol{z}_t$)
2: Compute sample estimates $\hat{\boldsymbol{A}}_y$ (Eq. 12) and $\hat{\boldsymbol{A}}_e$ (Eq. 13)
3: Set $\hat{\boldsymbol{B}} = \hat{\boldsymbol{A}}_e + \epsilon \boldsymbol{I}$
4: Solve generalized eigenvalue problem $\hat{\boldsymbol{A}}_y \boldsymbol{w} = \lambda \hat{\boldsymbol{B}} \boldsymbol{w}$ (Eq. 15)
5: Extract top $d_s$ generalized eigenvectors as $\boldsymbol{W}_s$ (with $\boldsymbol{W}_s^\top \hat{\boldsymbol{B}} \boldsymbol{W}_s = \boldsymbol{I}$)
6: Form $\boldsymbol{U}_s = (\boldsymbol{W}_s)$ and compute control limits for $T^2$ and SPE using training data $\boldsymbol{W}_s, \boldsymbol{U}_s$, control limits

### 5.2. Properties of Key Matrices

**Lemma 5.2** (Properties of $\boldsymbol{A}_y$ and $\boldsymbol{A}_e$)**.** *The causal signal matrix $\boldsymbol{A}_y$ defined in (12) is positive semidefinite with* $\text{rank}(\boldsymbol{A}_y) \leq L$, *and for any $\boldsymbol{w}$ we have $\boldsymbol{w}^\top \boldsymbol{A}_y \boldsymbol{w} = \sum_{i=1}^L (\boldsymbol{w}^\top \boldsymbol{\Sigma}_{zy_i})^2 / \sigma_{y_i}^2$. The environment sensitivity matrix $\boldsymbol{A}_e$ defined in (13) is positive semidefinite with $\text{rank}(\boldsymbol{A}_e) \leq K - 1$, and satisfies the law of total variance $\text{Var}(\boldsymbol{z}) = \mathbb{E}[\text{Var}(\boldsymbol{z} \mid e)] + \boldsymbol{A}_e$. Moreover, for any mean-invariant direction $\boldsymbol{w}$ satisfying $\boldsymbol{w}^\top \boldsymbol{\mu}_z^{(k)}$ constant over $k$, we have $\boldsymbol{w}^\top \boldsymbol{A}_e \boldsymbol{w} = 0$.*

*Proof Sketch.* For $\boldsymbol{A}_y$, expanding $\boldsymbol{w}^\top \boldsymbol{A}_y \boldsymbol{w}$ yields a sum of squared terms, establishing positive semidefiniteness; each summand is rank-1, so $\text{rank}(\boldsymbol{A}_y) \leq L$. For $\boldsymbol{A}_e$, positive semidefiniteness follows from its form as a weighted sum of outer products; the rank bound follows from the constraint $\sum_k \pi_k (\boldsymbol{\mu}_z^{(k)} - \bar{\boldsymbol{\mu}}_z) = \boldsymbol{0}$. See Appendix A.1 for details. □

### 5.3. Identification of Mean-Invariant Predictive Subspace

**Theorem 5.3** (Mean-Invariant Predictive Subspace Identification)**.** *Let $\boldsymbol{B} = \boldsymbol{A}_e + \epsilon \boldsymbol{I} \succ 0$ and suppose $^d = \mathcal{C} \oplus \mathcal{S}$ as in Assumption 5.1. Assume (i) environment diversity on $\mathcal{S}$: there exists $\sigma_s > 0$ such that $\boldsymbol{w}^\top \boldsymbol{A}_e \boldsymbol{w} \geq \sigma_s \|\boldsymbol{w}\|^2$ for all $\boldsymbol{w} \in \mathcal{S}$; and (ii) predictive richness on $\mathcal{C}$: there exists $\gamma_c > 0$ such that $\boldsymbol{w}^\top \boldsymbol{A}_y \boldsymbol{w} \geq \gamma_c \|\boldsymbol{w}\|^2$ for all $\boldsymbol{w} \in \mathcal{C} \setminus \{0\}$. These conditions implicitly require $\dim(\mathcal{S}) \leq K - 1$ and $\dim(\mathcal{C}) \leq L$. Then, as $\epsilon \to 0$, the top $r_c = \dim(\mathcal{C})$ generalized eigenvalues of $\boldsymbol{A}_y \boldsymbol{w} = \lambda(\boldsymbol{A}_e + \epsilon \boldsymbol{I}) \boldsymbol{w}$ diverge on the order of $1/\epsilon$, while the remaining eigenvalues stay bounded. The corresponding generalized eigenspace converges to $\mathcal{C}$.*

*Proof Sketch.* Define $R(\boldsymbol{w}) = \boldsymbol{w}^\top \boldsymbol{A}_y \boldsymbol{w} / \boldsymbol{w}^\top \boldsymbol{B} \boldsymbol{w}$. For $\boldsymbol{w} \in \mathcal{C} \setminus \{0\}$, Assumption 5.1 gives $\mathcal{C} \subseteq \text{ker}(\boldsymbol{A}_e)$, so $\boldsymbol{w}^\top \boldsymbol{A}_e \boldsymbol{w} = 0$ and $R(\boldsymbol{w}) \geq \gamma_c / \epsilon \to \infty$. For $\boldsymbol{w} \in \mathcal{S}$, environment diversity gives $R(\boldsymbol{w}) \leq \|\boldsymbol{A}_y\|_2 / \sigma_s$, which is bounded. The min–max theorem then implies spectral

separation: top $r_c$ eigenvalues diverge while others remain bounded. See Appendix A.2 for the complete argument. □

**Corollary 5.4** (Spectral Gap)**.** *Under the conditions of Theorem 5.3, for $\epsilon$ sufficiently small,*

$$\frac{\lambda_{r_c}}{\lambda_{r_c+1}} \geq \frac{\gamma_c \sigma_s}{\epsilon \|\boldsymbol{A}_y\|_2}. \tag{20}$$

*Proof Sketch.* From Theorem 5.3, $\lambda_{r_c} \geq \gamma_c / \epsilon$ and $\lambda_{r_c+1} \leq \|\boldsymbol{A}_y\|_2 / \sigma_s$. The ratio follows immediately. See Appendix A.3 for details. □

### 5.4. Finite-Sample Analysis

**Assumption 5.5** (Sub-Gaussian Samples)**.** *The centered observations $(\boldsymbol{z}_t - \mathbb{E}[\boldsymbol{z}_t], \boldsymbol{y}_t - \mathbb{E}[\boldsymbol{y}_t])$ are sub-Gaussian with parameter $\sigma^2$, and samples are i.i.d. across $t$ or satisfy a mixing condition with effective sample size $N_{\text{eff}}$.*

**Theorem 5.6** (Convergence Rate)**.** *Let $\hat{\boldsymbol{A}}_y$ and $\hat{\boldsymbol{A}}_e$ be sample estimates from $N$ observations, with $d = m(s + 1)$, $N_{\min} = \min_k N_k$, and $M = \max_k \|\boldsymbol{\mu}_z^{(k)} - \bar{\boldsymbol{\mu}}_z\|_2$. Under Assumption 5.5 with $\min_i \sigma_{y_i}^2 \geq c_y > 0$, with probability at least $1 - \delta$,*

$$\|\hat{\boldsymbol{A}}_y - \boldsymbol{A}_y\|_2 = O\left( \frac{\|\boldsymbol{\Sigma}_{zy}\|_2 \sigma^2}{c_y} \sqrt{\frac{d + L}{N}} \right.$$
$$\left. + \frac{\sigma^4}{c_y} \frac{d + L}{N} \right), \tag{21}$$

$$\|\hat{\boldsymbol{A}}_e - \boldsymbol{A}_e\|_2 = O\left( M\sigma \sqrt{\frac{d}{N_{\min}}} + \sigma^2 \frac{d}{N_{\min}} \right), \tag{22}$$

*where logarithmic factors in $1/\delta$ and $K$ are absorbed into the big-O notation.*

*Proof Sketch.* For $\boldsymbol{A}_y$, we decompose $\hat{\boldsymbol{A}}_y - \boldsymbol{A}_y$ into linear and quadratic terms in the cross-covariance estimation error, then apply matrix concentration bounds for sub-Gaussian random matrices. For $\boldsymbol{A}_e$, we use concentration of sample means per environment and apply a union bound over $K$ environments. The second term in each bound ensures non-trivial rates even when $\|\boldsymbol{\Sigma}_{zy}\|_2$ or $M$ is small. See Appendix A.4 for the complete derivation. □

**Corollary 5.7** (Subspace Estimation Error)**.** *Let $\hat{\boldsymbol{W}}_s$ and $\boldsymbol{W}_s^*$ be the CFL solutions from sample and population quantities, respectively. If the eigengap satisfies $\lambda_{d_s} - \lambda_{d_s+1} \geq \Delta > 0$ and $\|\hat{\boldsymbol{A}}_e - \boldsymbol{A}_e\|_2 \leq c\epsilon$ for some $c < 1$, then with high probability,*

$$\sin \angle(span(\hat{\boldsymbol{W}}_s), span(\boldsymbol{W}_s^*))$$
$$= O\left( \frac{1}{\Delta \epsilon} \left( \|\hat{\boldsymbol{A}}_y - \boldsymbol{A}_y\|_2 + \|\hat{\boldsymbol{A}}_e - \boldsymbol{A}_e\|_2 \right) \right). \tag{23}$$

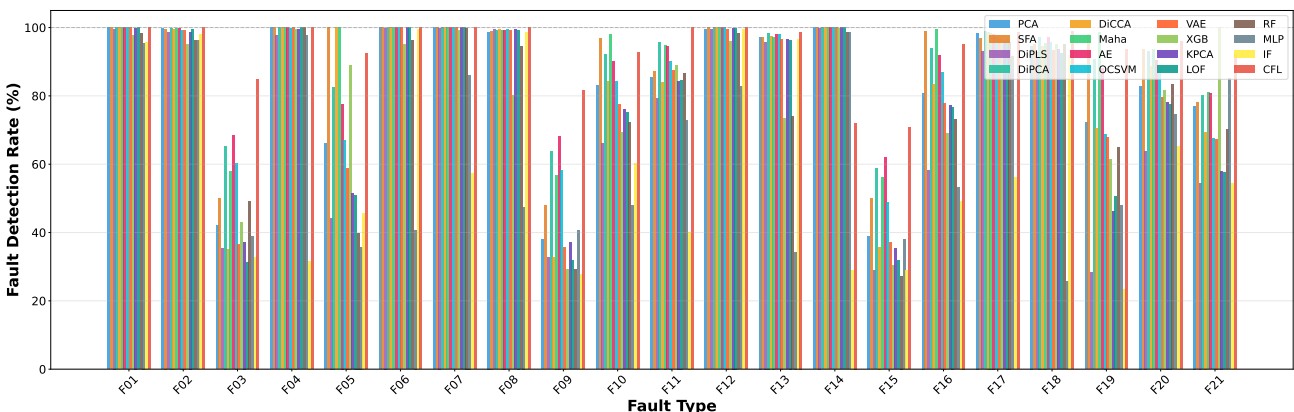

*Figure 3.* Per-fault FDR comparison across all 16 methods on the Tennessee Eastman Process. CFL achieves the highest average FDR (93.7%), with particularly large improvements on difficult faults (Faults 03, 09, 15, 21) that involve subtle process changes requiring both temporal dynamics and invariance for reliable detection.

*Proof Sketch.* Generalized eigenvectors of $(\boldsymbol{A}_y, \boldsymbol{B})$ correspond to eigenvectors of $\boldsymbol{M} = \boldsymbol{B}^{-1/2} \boldsymbol{A}_y \boldsymbol{B}^{-1/2}$. The Davis–Kahan theorem bounds subspace perturbation by $\|\hat{\boldsymbol{M}} - \boldsymbol{M}\|_2 / \Delta$, and the condition $\|\hat{\boldsymbol{A}}_e - \boldsymbol{A}_e\|_2 \leq c\epsilon$ ensures stability of the whitening with $\lambda_{\min}(\boldsymbol{B}) \geq \epsilon$. See Appendix A.5 for details. $\quad\square$

### 5.5. Connection to Causal Inference

**Proposition 5.8** (Temporal Relevance via Lagged Cross–Covariance)**.** *Let $\boldsymbol{e}_{j,k} \in^d$ be the canonical basis vector corresponding to $X_{j,t-k}$ in $\boldsymbol{z}_t$. If $(X_{j,t-k}, y_{i,t}) \neq 0$ for some target $y_i$, then $\boldsymbol{e}_{j,k}^\top \boldsymbol{A}_y \boldsymbol{e}_{j,k} > 0$.*

*Proof Sketch.* A nonzero covariance implies $[\boldsymbol{\Sigma}_{zy_i}]_{j+km} \neq 0$. Since $\boldsymbol{e}_{j,k}^\top \boldsymbol{A}_y \boldsymbol{e}_{j,k} = \sum_{\ell=1}^L [\boldsymbol{\Sigma}_{zy_\ell}]_{j+km}^2 / \sigma_{y_\ell}^2$, the term for $\ell = i$ is strictly positive. See Appendix A.6 for details. $\quad\square$

**Proposition 5.9** (Invariance of the Conditional Mechanism)**.** *Assume a structural equation model $\boldsymbol{Y} = f(\boldsymbol{z}_c) + \boldsymbol{\eta}$ holds across environments, where $f$ and the distribution of $\boldsymbol{\eta}$ do not depend on $e$, and $(\boldsymbol{z}_c, \boldsymbol{\eta})e$. If $\boldsymbol{s} = \boldsymbol{W}_s^\top \boldsymbol{z}$ spans $\mathcal{C}$ and $\boldsymbol{W}_s$ is injective on $\mathcal{C}$, then $P^e(\boldsymbol{Y} \mid \boldsymbol{s}) = P^{e'}(\boldsymbol{Y} \mid \boldsymbol{s})$ for all $e, e' \in \mathcal{E}$.*

*Proof Sketch.* Since $\boldsymbol{s}$ is an invertible representation of $\boldsymbol{z}_c$ on $\mathcal{C}$, there exists $g$ such that $\boldsymbol{z}_c = g(\boldsymbol{s})$. Thus $\boldsymbol{Y} \mid \boldsymbol{s} \stackrel{d}{=} f(g(\boldsymbol{s})) + \boldsymbol{\eta}$. The environment-invariance of $f$ and $P(\boldsymbol{\eta})$, together with $(\boldsymbol{z}_c, \boldsymbol{\eta})e$, implies $P^e(\boldsymbol{Y} \mid \boldsymbol{s})$ is independent of $e$. See Appendix A.7 for details. $\quad\square$

**Proposition 5.10** (Complexity Analysis)**.** *The computational complexity of CFL is $O(Nd^2 + d^3)$, where $d = m(s+1)$. For $N \gg d$, the cost is dominated by covariance estimation.*

*Proof Sketch.* Time-lagged matrix construction requires $O(Nms)$, covariance estimation requires $O(Nd^2)$, and the generalized eigenvalue problem requires $O(d^3)$. Summing these yields the total complexity. See Appendix A.8 for the breakdown. $\quad\square$

## 6. Experiments

### 6.1. Experimental Setup

We evaluate CFL on the Tennessee Eastman Process (TEP) (Downs & Vogel, 1993). The Tennessee Eastman Process is a widely-used benchmark that simulates a chemical plant with 52 variables, comprising 41 process measurements (XMEAS 1–41) and 11 manipulated variables (XMV 1–11). The process involves interconnected reactor, separator, and stripper units with known causal relationships determined by material and energy flows. We use the extended dataset (Reinartz et al., 2021) containing 21 distinct fault types, where training data consists of 480 normal-operation samples and test data contains 960 samples with faults introduced after sample 160.

For target variables that define the causal signal matrix $\boldsymbol{A}_y$, we select process measurements reflecting key downstream states that are expected to be causally influenced by upstream disturbances. This selection ensures that the extracted features capture information propagating through the main causal pathways of the process.

We compare CFL against 15 baseline methods spanning two categories. The first category comprises dynamic latent variable methods from the process monitoring literature: PCA (Jackson, 1991) as a static baseline, SFA (Wiskott & Sejnowski, 2002) which extracts slowly varying features, DiPLS (Y.Dong & Qin, 2015) which incorporates output variables, DiPCA (Ku et al., 1995) which models autoregressive dynamics, and DiCCA (Dong & Qin, 2018) which

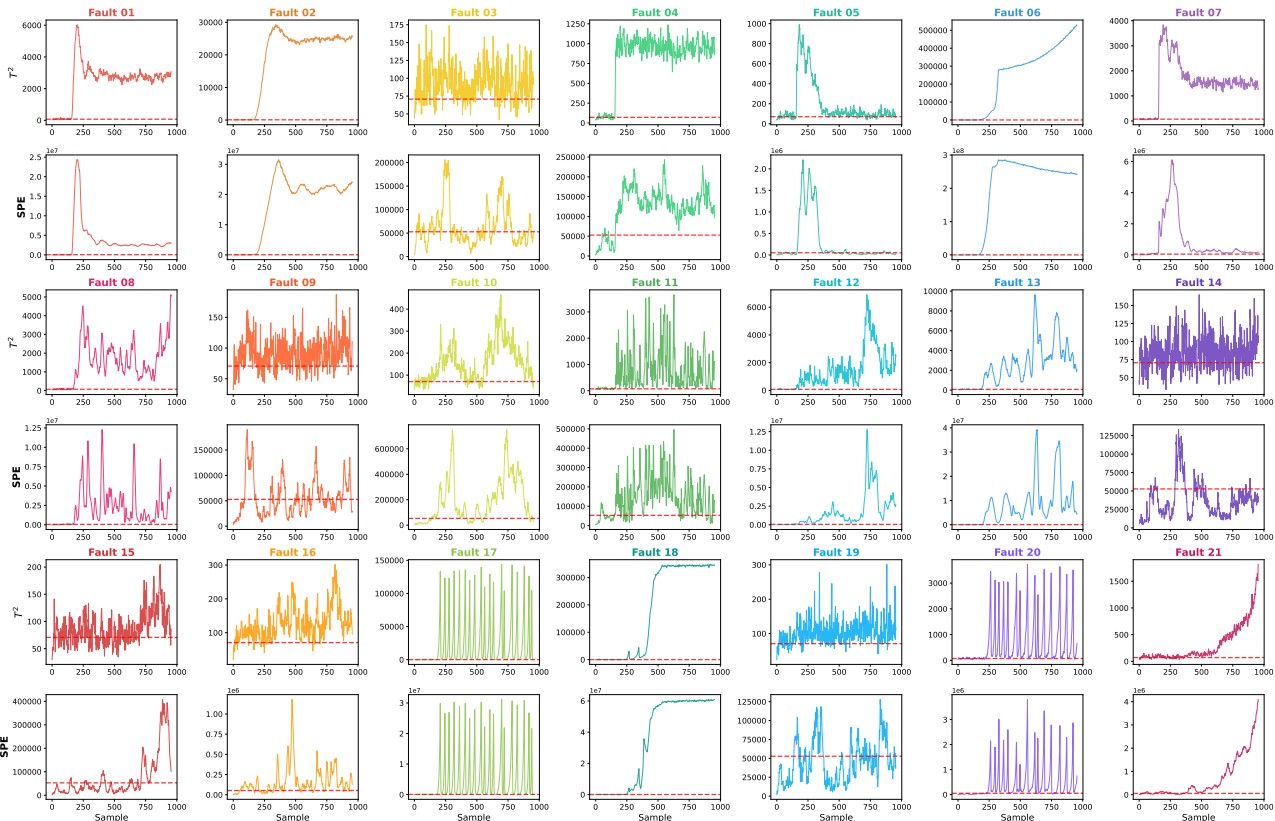

*Figure 4.* CFL fault detection monitoring results for all 21 TEP faults. Red dashed lines indicate control limits at 95% significance level. CFL detects faults after the fault introduction point (sample 160), with clear exceedances above the control limits for most fault types.

maximizes canonical correlations with lagged variables.

The second category includes ten deep learning and anomaly detection methods: Mahalanobis distance (Mahalanobis, 1936) which measures deviation from normal data distribution, autoencoder (Hinton & Salakhutdinov, 2006) and variational autoencoder (VAE) (Kingma & Welling, 2014) using reconstruction error for anomaly detection, one-class SVM (OCSVM) (Schölkopf et al., 2001) which learns a decision boundary around normal data, isolation forest (Liu et al., 2008) which isolates anomalies through random partitioning, local outlier factor (LOF) (Breunig et al., 2000) which detects anomalies based on local density deviation, Kernel PCA (Schölkopf et al., 1998) using nonlinear reconstruction error, and supervised ensemble methods including random forest (Breiman, 2001), XGBoost (Chen & Guestrin, 2016), and multi-layer perceptron (MLP) (Rumelhart et al., 1986) trained with pseudo-labels. For CFL, we set variance threshold 0.9, significance level 0.95, lag order $s = 10$, and $\epsilon = 5 \times 10^{-4}$.

### 6.2. Main Results

As shown in Table 1 and Figure 3, CFL achieves 93.7% average fault detection rate, outperforming all 15 baseline

methods spanning dynamic latent variable approaches, deep learning techniques, and classical anomaly detection algorithms. The performance gap widens substantially for methods that lack either temporal modeling or robustness to distribution shifts: Isolation Forest achieves only 61.0% and MLP reaches 63.4%, demonstrating the importance of jointly capturing both aspects that CFL optimizes.

The performance gains are particularly pronounced on difficult faults that involve subtle process changes and operating variability. On Faults 03, 09, 15, and 21, which are notoriously challenging due to their small magnitude or slow development, CFL achieves detection rates of 84.9%, 81.6%, 70.8%, and 92.1% respectively, substantially exceeding all baselines. As illustrated in Figure 4, the $T^2$ and SPE statistics exhibit clear exceedances above control limits for these difficult faults, whereas baseline methods often fail to trigger reliable alarms. These faults require both temporal dynamic information to capture fault propagation patterns and robustness to normal operating variations, which CFL provides through its joint optimization of $\boldsymbol{A}_y$ and $\boldsymbol{A}_e$.

*Table 1.* Fault Detection Rate (%) for all 21 TEP faults across 16 methods. Green indicates the best result for each fault.

| Fault | PCA | SFA | DiPLS | DiPCA | DiCCA | Maha | AE | OCSVM | VAE | XGB | KPCA | LOF | RF | MLP | IF | CFL |
|---|---|---|---|---|---|---|---|---|---|---|---|---|---|---|---|---|
| F01 | 100.0 | 100.0 | 99.6 | 100.0 | 100.0 | 100.0 | 100.0 | 100.0 | 100.0 | 97.8 | 99.9 | 100.0 | 98.3 | 95.5 | 95.8 | 100.0 |
| F02 | 99.8 | 99.6 | 98.6 | 99.9 | 99.6 | 99.8 | 99.8 | 99.3 | 99.3 | 95.0 | 98.8 | 99.4 | 96.3 | 96.3 | 98.0 | 100.0 |
| F03 | 42.0 | 50.0 | 35.4 | 65.3 | 35.1 | 57.9 | 68.4 | 60.4 | 36.5 | 43.0 | 37.3 | 31.3 | 49.0 | 38.8 | 32.8 | 84.9 |
| F04 | 100.0 | 100.0 | 97.6 | 100.0 | 100.0 | 100.0 | 100.0 | 99.9 | 100.0 | 99.6 | 99.4 | 100.0 | 100.0 | 97.8 | 31.6 | 100.0 |
| F05 | 66.1 | 100.0 | 44.3 | 82.6 | 100.0 | 100.0 | 77.6 | 66.9 | 58.9 | 89.0 | 51.4 | 51.0 | 39.9 | 35.8 | 45.8 | 92.3 |
| F06 | 100.0 | 100.0 | 99.9 | 100.0 | 100.0 | 100.0 | 100.0 | 100.0 | 100.0 | 95.0 | 100.0 | 100.0 | 96.1 | 40.8 | 99.5 | 100.0 |
| F07 | 100.0 | 100.0 | 99.9 | 100.0 | 100.0 | 100.0 | 100.0 | 100.0 | 100.0 | 99.1 | 100.0 | 100.0 | 99.9 | 86.1 | 57.3 | 100.0 |
| F08 | 98.8 | 99.0 | 99.4 | 99.1 | 99.4 | 99.3 | 99.3 | 99.5 | 99.1 | 80.1 | 99.4 | 99.3 | 94.5 | 47.5 | 98.6 | 100.0 |
| F09 | 37.9 | 47.9 | 32.8 | 63.8 | 32.7 | 56.9 | 68.3 | 58.1 | 35.8 | 29.1 | 37.1 | 31.9 | 29.3 | 40.5 | 27.9 | 81.6 |
| F10 | 83.1 | 96.9 | 66.3 | 92.2 | 84.2 | 97.9 | 90.0 | 84.1 | 77.5 | 69.3 | 76.0 | 75.1 | 72.1 | 47.9 | 60.4 | 92.7 |
| F11 | 85.5 | 87.3 | 79.4 | 95.7 | 83.9 | 94.9 | 94.4 | 90.3 | 87.5 | 88.9 | 84.4 | 84.6 | 86.6 | 72.8 | 40.1 | 100.0 |
| F12 | 99.4 | 100.0 | 99.4 | 100.0 | 100.0 | 100.0 | 100.0 | 100.0 | 99.6 | 96.0 | 99.9 | 99.9 | 98.3 | 82.8 | 99.4 | 100.0 |
| F13 | 97.3 | 97.1 | 95.8 | 98.4 | 97.4 | 97.3 | 98.0 | 97.9 | 96.6 | 73.4 | 96.5 | 96.4 | 74.0 | 34.3 | 96.5 | 98.7 |
| F14 | 100.0 | 100.0 | 99.9 | 100.0 | 100.0 | 100.0 | 100.0 | 100.0 | 100.0 | 99.8 | 100.0 | 100.0 | 98.5 | 98.5 | 29.0 | 72.1 |
| F15 | 38.9 | 50.0 | 28.9 | 58.7 | 35.7 | 56.1 | 61.9 | 48.9 | 37.0 | 30.4 | 35.3 | 31.9 | 27.3 | 38.0 | 28.9 | 70.8 |
| F16 | 80.9 | 98.9 | 58.3 | 93.8 | 83.4 | 99.4 | 91.9 | 87.0 | 77.9 | 69.0 | 77.1 | 76.8 | 73.0 | 53.3 | 49.3 | 95.2 |
| F17 | 98.4 | 96.8 | 93.0 | 98.9 | 98.7 | 98.4 | 98.1 | 97.0 | 96.5 | 96.3 | 95.9 | 96.6 | 95.8 | 91.5 | 56.1 | 98.6 |
| F18 | 94.6 | 95.0 | 92.8 | 97.2 | 94.6 | 95.5 | 97.1 | 95.6 | 93.3 | 95.0 | 93.6 | 92.5 | 95.0 | 25.6 | 91.8 | 98.9 |
| F19 | 72.1 | 98.0 | 28.4 | 90.7 | 70.4 | 98.5 | 88.0 | 68.9 | 67.8 | 61.4 | 46.3 | 50.5 | 64.9 | 47.9 | 23.3 | 93.6 |
| F20 | 82.9 | 93.5 | 63.9 | 93.0 | 88.7 | 93.6 | 90.4 | 87.4 | 79.5 | 81.8 | 78.3 | 77.6 | 83.4 | 74.6 | 65.1 | 96.0 |
| F21 | 77.0 | 78.1 | 54.5 | 80.1 | 69.4 | 81.0 | 80.6 | 67.6 | 67.4 | 100.0 | 58.0 | 57.5 | 70.3 | 85.1 | 54.5 | 92.1 |
| Avg | 83.6 | 89.9 | 74.7 | 90.9 | 84.4 | 91.7 | 90.7 | 86.1 | 81.4 | 80.4 | 79.3 | 78.7 | 78.2 | 63.4 | 61.0 | **93.7** |

*Table 2.* Ablation study of CFL hyperparameters on average FDR.

| $s$ | 2 | 4 | 6 | 8 | 10 |
|---|---|---|---|---|---|
| FDR | 68.7 | 76.7 | 83.6 | 89.2 | **93.7** |
| $\epsilon$ | $5\times10^{-4}$ | $10^{-3}$ | $2\times10^{-3}$ | $5\times10^{-3}$ | $10^{-2}$ |
| FDR | **93.7** | 92.5 | 90.9 | 88.2 | 85.1 |
| Threshold | 0.70 | 0.80 | 0.85 | 0.88 | 0.90 |
| FDR | 69.8 | 76.8 | 87.3 | 92.1 | **93.7** |

## 6.3. Ablation Studies

We conduct ablation studies to analyze the sensitivity of CFL to its key hyperparameters, varying each parameter while holding others at default values (lag order $s = 10$, regularization $\epsilon = 5 \times 10^{-4}$, threshold= 0.9).

Table 2 presents the sensitivity analysis. For lag order $s$, performance increases monotonically from 68.7% at $s = 2$ to 93.7% at $s = 10$, confirming that sufficient temporal depth is necessary to capture fault propagation dynamics. For regularization $\epsilon$, the optimal value $\epsilon = 5 \times 10^{-4}$ balances emphasis on $\Sigma_d$ against temporal smoothness; larger values reduce detection capability. The variance threshold determines the effective latent dimension $a$, with performance improving from 69.8% at threshold 0.70 to 93.7% at threshold 0.90.

## 7. Conclusion

This work addressed the challenge of extracting robust causal features from time-series data under distribution shifts. We found that jointly optimizing temporal target relevance and environment mean invariance yields monitoring features that outperform methods focusing on only one aspect. CFL provides a principled spectral framework based on generalized Rayleigh quotients, with theoretical guarantees for identifying mean-invariant predictive subspaces. The results demonstrate that incorporating causal invariance principles into feature learning can significantly improve robustness in industrial process monitoring. Future work includes automatic target discovery, stronger distributional invariance via kernel-based penalties, and nonlinear extensions using neural networks.

## Impact Statement

This paper presents work whose goal is to advance the field of Machine Learning, and more specifically, the theoretical foundations of combining temporal causality and invariance principles in feature learning. There are many potential societal consequences of our work, none which we feel must be specifically highlighted here.

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

# A. Detailed Proofs of Theoretical Results

## A.1. Proof of Lemma 5.2

**Properties of $A_y$.**

(i) *Positive semidefiniteness.* For any $\boldsymbol{w} \in^d$,

$$\boldsymbol{w}^\top \boldsymbol{A}_y \boldsymbol{w} = \boldsymbol{w}^\top \left( \sum_{i=1}^{L} \boldsymbol{\Sigma}_{zy_i} \sigma_{y_i}^{-2} \boldsymbol{\Sigma}_{y_i z} \right) \boldsymbol{w} = \sum_{i=1}^{L} \frac{(\boldsymbol{w}^\top \boldsymbol{\Sigma}_{zy_i})^2}{\sigma_{y_i}^2} \geq 0.$$

The last equality uses $\boldsymbol{\Sigma}_{y_i z} = \boldsymbol{\Sigma}_{zy_i}^\top$ and the fact that $\boldsymbol{\Sigma}_{zy_i} \in^d$ is a vector, so $\boldsymbol{w}^\top \boldsymbol{\Sigma}_{zy_i}$ is a scalar.

(ii) *Rank bound.* Each term $\boldsymbol{\Sigma}_{zy_i} \sigma_{y_i}^{-2} \boldsymbol{\Sigma}_{y_i z} = \sigma_{y_i}^{-2} \boldsymbol{\Sigma}_{zy_i} \boldsymbol{\Sigma}_{zy_i}^\top$ is a rank-1 matrix (outer product of $\boldsymbol{\Sigma}_{zy_i}$ with itself). Since $\boldsymbol{A}_y$ is a sum of $L$ rank-1 matrices, $\mathrm{rank}(\boldsymbol{A}_y) \leq L$.

(iii) *Trace formula.* We have

$$(\boldsymbol{A}_y) = \sum_{i=1}^{L} \sigma_{y_i}^{-2} (\boldsymbol{\Sigma}_{zy_i} \boldsymbol{\Sigma}_{y_i z}) = \sum_{i=1}^{L} \sigma_{y_i}^{-2} \|\boldsymbol{\Sigma}_{zy_i}\|_2^2 = \|\boldsymbol{\Sigma}_{zy} \boldsymbol{D}_y^{-1/2}\|_F^2,$$

where $\boldsymbol{D}_y = \mathrm{diag}(\sigma_{y_1}^2, \dots, \sigma_{y_L}^2)$ and $\boldsymbol{\Sigma}_{zy} = [\boldsymbol{\Sigma}_{zy_1}, \dots, \boldsymbol{\Sigma}_{zy_L}] \in^{d \times L}$.

If $(\boldsymbol{z}) = \boldsymbol{I}$, then the coefficient of determination for regressing $y_i$ on $\boldsymbol{z}$ is

$$R^2(y_i \mid \boldsymbol{z}) = \frac{\boldsymbol{\Sigma}_{y_i z}(\boldsymbol{z})^{-1} \boldsymbol{\Sigma}_{zy_i}}{\sigma_{y_i}^2} = \frac{\|\boldsymbol{\Sigma}_{zy_i}\|_2^2}{\sigma_{y_i}^2},$$

so $(\boldsymbol{A}_y) = \sum_{i=1}^{L} R^2(y_i \mid \boldsymbol{z})$.

**Properties of $A_e$.**

(i) *Positive semidefiniteness.* The matrix $\boldsymbol{A}_e = \sum_{k=1}^{K} \pi_k (\boldsymbol{\mu}_z^{(k)} - \bar{\boldsymbol{\mu}}_z)(\boldsymbol{\mu}_z^{(k)} - \bar{\boldsymbol{\mu}}_z)^\top$ is a weighted sum of outer products, hence positive semidefinite.

(ii) *Rank bound.* Since $\sum_{k=1}^{K} \pi_k (\boldsymbol{\mu}_z^{(k)} - \bar{\boldsymbol{\mu}}_z) = \bar{\boldsymbol{\mu}}_z - \bar{\boldsymbol{\mu}}_z = \boldsymbol{0}$, the $K$ centered mean vectors satisfy one linear constraint and thus span at most a $(K - 1)$-dimensional subspace. Hence $\mathrm{rank}(\boldsymbol{A}_e) \leq K - 1$.

(iii) *Law of total variance.* Let $e$ be a random environment with $(e = e_k) = \pi_k$. Then $\boldsymbol{A}_e = \mathrm{Var}(\mathbb{E}[\boldsymbol{z} \mid e])$, and the law of total variance gives

$$\mathrm{Var}(\boldsymbol{z}) = \mathbb{E}[\mathrm{Var}(\boldsymbol{z} \mid e)] + \mathrm{Var}(\mathbb{E}[\boldsymbol{z} \mid e]) = \mathbb{E}[\mathrm{Var}(\boldsymbol{z} \mid e)] + \boldsymbol{A}_e.$$

(iv) *Mean-invariant directions.* If $\boldsymbol{w}^\top \boldsymbol{\mu}_z^{(k)} = c$ (constant) for all $k$, then $\boldsymbol{w}^\top \bar{\boldsymbol{\mu}}_z = \sum_k \pi_k \boldsymbol{w}^\top \boldsymbol{\mu}_z^{(k)} = c$. Thus $\boldsymbol{w}^\top (\boldsymbol{\mu}_z^{(k)} - \bar{\boldsymbol{\mu}}_z) = 0$ for all $k$, and

$$\boldsymbol{w}^\top \boldsymbol{A}_e \boldsymbol{w} = \sum_{k=1}^{K} \pi_k (\boldsymbol{w}^\top (\boldsymbol{\mu}_z^{(k)} - \bar{\boldsymbol{\mu}}_z))^2 = 0.$$

## A.2. Proof of Theorem 5.3

Define the generalized Rayleigh quotient

$$R(\boldsymbol{w}) = \frac{\boldsymbol{w}^\top \boldsymbol{A}_y \boldsymbol{w}}{\boldsymbol{w}^\top (\boldsymbol{A}_e + \epsilon \boldsymbol{I}) \boldsymbol{w}}.$$

**Step 1: Lower bound on $\mathcal{C}$.** For any $\boldsymbol{w} \in \mathcal{C} \setminus \{0\}$, Assumption 5.1 states that $\mathcal{C} \subseteq \mathrm{ker}(\boldsymbol{A}_e)$, which directly implies $\boldsymbol{w}^\top \boldsymbol{A}_e \boldsymbol{w} = 0$. By the predictive richness condition, $\boldsymbol{w}^\top \boldsymbol{A}_y \boldsymbol{w} \geq \gamma_c \|\boldsymbol{w}\|^2$. Therefore,

$$R(\boldsymbol{w}) = \frac{\boldsymbol{w}^\top \boldsymbol{A}_y \boldsymbol{w}}{\boldsymbol{w}^\top (\boldsymbol{A}_e + \epsilon \boldsymbol{I}) \boldsymbol{w}} = \frac{\boldsymbol{w}^\top \boldsymbol{A}_y \boldsymbol{w}}{\epsilon \|\boldsymbol{w}\|^2} \geq \frac{\gamma_c}{\epsilon} \to \infty \quad (\epsilon \to 0).$$

**Step 2: Upper bound on $\mathcal{S}$.** For any $\boldsymbol{w} \in \mathcal{S} \setminus \{0\}$, the environment diversity condition gives $\boldsymbol{w}^\top \boldsymbol{A}_e \boldsymbol{w} \geq \sigma_s \|\boldsymbol{w}\|^2$. Also, $\boldsymbol{w}^\top \boldsymbol{A}_y \boldsymbol{w} \leq \|\boldsymbol{A}_y\|_2 \|\boldsymbol{w}\|^2$. Thus,

$$R(\boldsymbol{w}) = \frac{\boldsymbol{w}^\top \boldsymbol{A}_y \boldsymbol{w}}{\boldsymbol{w}^\top (\boldsymbol{A}_e + \epsilon \boldsymbol{I}) \boldsymbol{w}} \leq \frac{\|\boldsymbol{A}_y\|_2 \|\boldsymbol{w}\|^2}{\sigma_s \|\boldsymbol{w}\|^2} = \frac{\|\boldsymbol{A}_y\|_2}{\sigma_s},$$

which is bounded uniformly in $\epsilon$.

**Step 3: Spectral separation via min–max.** By the Courant–Fischer min–max theorem for generalized eigenvalues,

$$\lambda_j = \max_{\dim(V)=j} \min_{\boldsymbol{w} \in V \setminus \{0\}} R(\boldsymbol{w}).$$

For $j \leq r_c = \dim(\mathcal{C})$, we can choose a $j$-dimensional subspace $V \subseteq \mathcal{C}$. For any $\boldsymbol{w} \in V \setminus \{0\}$, Step 1 gives $R(\boldsymbol{w}) \geq \gamma_c / \epsilon$. Thus $\lambda_j \geq \gamma_c / \epsilon$ for $j \leq r_c$.

For $j = r_c + 1$, any $(r_c + 1)$-dimensional subspace $V$ must satisfy $V \cap \mathcal{S} \neq \{0\}$ (since $\dim(\mathcal{C}) = r_c$). Let $\boldsymbol{w} \in V \cap \mathcal{S} \setminus \{0\}$. By Step 2, $R(\boldsymbol{w}) \leq \|\boldsymbol{A}_y\|_2 / \sigma_s$. Thus $\lambda_{r_c+1} \leq \|\boldsymbol{A}_y\|_2 / \sigma_s$.

As $\epsilon \to 0$, the top $r_c$ eigenvalues diverge like $1/\epsilon$ while $\lambda_{r_c+1}$ remains bounded. For the corresponding eigenvectors, any $\mathcal{S}$-component would contribute a bounded Rayleigh quotient; the divergence of eigenvalues forces the $\mathcal{S}$-component to vanish as $\epsilon \to 0$, so the eigenspace converges to $\mathcal{C}$.

### A.3. Proof of Corollary 5.4

From Step 1 of Theorem 5.3, $\lambda_{r_c} \geq \gamma_c / \epsilon$.

From Step 2 and the min–max characterization, $\lambda_{r_c+1} \leq \|\boldsymbol{A}_y\|_2 / \sigma_s$.

Taking the ratio:

$$\frac{\lambda_{r_c}}{\lambda_{r_c+1}} \geq \frac{\gamma_c / \epsilon}{\|\boldsymbol{A}_y\|_2 / \sigma_s} = \frac{\gamma_c \sigma_s}{\epsilon \|\boldsymbol{A}_y\|_2}.$$

### A.4. Proof of Theorem 5.6

**Preliminary: Concentration of Cross-Covariance.**

**Lemma A.1** (Operator-Norm Concentration). *Under Assumption 5.5, with probability at least $1 - \delta$:*

$$\|\hat{\boldsymbol{\Sigma}}_{zy} - \boldsymbol{\Sigma}_{zy}\|_2 \leq C\sigma^2 \sqrt{\frac{d + L + \log(2/\delta)}{N}}.$$

*Proof.* This follows from standard matrix concentration bounds for sub-Gaussian random matrices. The sample cross-covariance $\hat{\boldsymbol{\Sigma}}_{zy} = \frac{1}{N} \sum_{t=1}^{N} (\boldsymbol{z}_t - \bar{\boldsymbol{z}})(\boldsymbol{y}_t - \bar{\boldsymbol{y}})^\top$ is a sum of independent rank-1 matrices (up to mean estimation error). Applying the non-commutative Bernstein inequality or $\epsilon$-net arguments yields the stated bound. $\square$

**Concentration of $\hat{\boldsymbol{A}}_y$.**

Recall $\boldsymbol{A}_y = \sum_{i=1}^{L} \boldsymbol{\Sigma}_{zy_i} \sigma_{y_i}^{-2} \boldsymbol{\Sigma}_{y_i z}$. Let $\boldsymbol{E}_i = \hat{\boldsymbol{\Sigma}}_{zy_i} - \boldsymbol{\Sigma}_{zy_i}$. Then

$$\hat{\boldsymbol{A}}_y - \boldsymbol{A}_y = \sum_{i=1}^{L} \left[ \hat{\boldsymbol{\Sigma}}_{zy_i} \hat{\sigma}_{y_i}^{-2} \hat{\boldsymbol{\Sigma}}_{y_i z} - \boldsymbol{\Sigma}_{zy_i} \sigma_{y_i}^{-2} \boldsymbol{\Sigma}_{y_i z} \right]$$

$$= \sum_{i=1}^{L} \sigma_{y_i}^{-2} \left[ \boldsymbol{\Sigma}_{zy_i} \boldsymbol{E}_i^\top + \boldsymbol{E}_i \boldsymbol{\Sigma}_{y_i z} + \boldsymbol{E}_i \boldsymbol{E}_i^\top \right]$$

$$+ \text{(variance estimation terms)}.$$

Using triangle inequality and $\|\boldsymbol{E}_i\|_2 \leq \|\hat{\boldsymbol{\Sigma}}_{zy} - \boldsymbol{\Sigma}_{zy}\|_2$:

$$\|\hat{\boldsymbol{A}}_y - \boldsymbol{A}_y\|_2 \leq \frac{2}{c_y} \|\boldsymbol{\Sigma}_{zy}\|_2 \|\hat{\boldsymbol{\Sigma}}_{zy} - \boldsymbol{\Sigma}_{zy}\|_2$$
$$+ \frac{1}{c_y} \|\hat{\boldsymbol{\Sigma}}_{zy} - \boldsymbol{\Sigma}_{zy}\|_2^2 + \text{(variance terms)}.$$

Substituting Lemma A.1 and noting that variance estimation contributes the same order under sub-Gaussian concentration yields the linear term $O(\|\boldsymbol{\Sigma}_{zy}\|_2 \sigma^2 \sqrt{(d+L)/N}/c_y)$ and the quadratic term $O(\sigma^4(d+L)/(c_y N))$.

**Concentration of $\hat{\boldsymbol{A}}_e$.**

**Lemma A.2** (Mean Concentration per Environment). *Under Assumption 5.5, for environment $k$ with $N_k$ samples, with probability at least $1 - \delta$:*

$$\|\hat{\boldsymbol{\mu}}_z^{(k)} - \boldsymbol{\mu}_z^{(k)}\|_2 \leq C\sigma \sqrt{\frac{d + \log(2/\delta)}{N_k}}.$$

Let $\boldsymbol{\delta}_k = \hat{\boldsymbol{\mu}}_z^{(k)} - \boldsymbol{\mu}_z^{(k)}$ and $\bar{\boldsymbol{\delta}} = \sum_k \pi_k \boldsymbol{\delta}_k$. Expanding:

$$\hat{\boldsymbol{A}}_e - \boldsymbol{A}_e$$
$$= \sum_{k=1}^{K} \pi_k \Big[ (\boldsymbol{\mu}_z^{(k)} - \bar{\boldsymbol{\mu}}_z + \boldsymbol{\delta}_k - \bar{\boldsymbol{\delta}})(\boldsymbol{\mu}_z^{(k)} - \bar{\boldsymbol{\mu}}_z + \boldsymbol{\delta}_k - \bar{\boldsymbol{\delta}})^\top$$
$$- (\boldsymbol{\mu}_z^{(k)} - \bar{\boldsymbol{\mu}}_z)(\boldsymbol{\mu}_z^{(k)} - \bar{\boldsymbol{\mu}}_z)^\top \Big].$$

The linear terms contribute $O(M \max_k \|\boldsymbol{\delta}_k\|_2)$ and quadratic terms contribute $O(\max_k \|\boldsymbol{\delta}_k\|_2^2)$. Applying Lemma A.2 with a union bound over $K$ environments:

$$\|\hat{\boldsymbol{A}}_e - \boldsymbol{A}_e\|_2 \leq C_2 \left( M\sigma \sqrt{\frac{d + \log(K/\delta)}{N_{\min}}} + \sigma^2 \frac{d + \log(K/\delta)}{N_{\min}} \right).$$

The second term ensures the bound remains meaningful even when $M = 0$.

### A.5. Proof of Corollary 5.7

Generalized eigenvectors of $(\boldsymbol{A}_y, \boldsymbol{B})$ correspond to standard eigenvectors of the whitened matrix

$$\boldsymbol{M} = \boldsymbol{B}^{-1/2} \boldsymbol{A}_y \boldsymbol{B}^{-1/2}.$$

Let $\hat{\boldsymbol{M}} = \hat{\boldsymbol{B}}^{-1/2} \hat{\boldsymbol{A}}_y \hat{\boldsymbol{B}}^{-1/2}$. By the Davis–Kahan sin $\Theta$ theorem, if $\lambda_{d_s}(\boldsymbol{M}) - \lambda_{d_s+1}(\boldsymbol{M}) \geq \Delta > 0$, then

$$\sin \angle(\text{span}(\hat{\boldsymbol{V}}_{d_s}), \text{span}(\boldsymbol{V}_{d_s}^*)) \leq \frac{\|\hat{\boldsymbol{M}} - \boldsymbol{M}\|_2}{\Delta},$$

where $\boldsymbol{V}_{d_s}$ denotes the top $d_s$ eigenvectors.

To bound $\|\hat{\boldsymbol{M}} - \boldsymbol{M}\|_2$, the condition $\|\hat{\boldsymbol{A}}_e - \boldsymbol{A}_e\|_2 \leq c\epsilon$ with $c < 1$ ensures that $\|\hat{\boldsymbol{B}} - \boldsymbol{B}\|_2 \leq c\epsilon$ and hence $\lambda_{\min}(\hat{\boldsymbol{B}}) \geq (1-c)\epsilon > 0$. This guarantees $\|\boldsymbol{B}^{-1/2}\|_2 \leq 1/\sqrt{\epsilon}$ and similarly for $\hat{\boldsymbol{B}}^{-1/2}$. Using perturbation bounds for matrix products:

$$\|\hat{\boldsymbol{M}} - \boldsymbol{M}\|_2 \leq \|\boldsymbol{B}^{-1/2}\|_2^2 \|\hat{\boldsymbol{A}}_y - \boldsymbol{A}_y\|_2$$
$$+ \text{(terms involving } \hat{\boldsymbol{B}} - \boldsymbol{B})$$
$$\leq \frac{C}{\epsilon} \left( \|\hat{\boldsymbol{A}}_y - \boldsymbol{A}_y\|_2 + \|\hat{\boldsymbol{A}}_e - \boldsymbol{A}_e\|_2 \right),$$

where $C$ depends on $\|\boldsymbol{A}_y\|_2$, $\|\boldsymbol{A}_e\|_2$, and $c$. Combining with Theorem 5.6 yields the stated bound.

## A.6. Proof of Proposition 5.8

If $(X_{j,t-k}, y_{i,t}) \neq 0$, then the corresponding entry of $\mathbf{\Sigma}_{zy_i}$ is nonzero. Specifically, since $\mathbf{z}_t = [\mathbf{x}_t^\top, \mathbf{x}_{t-1}^\top, \ldots, \mathbf{x}_{t-s}^\top]^\top$, the coordinate $X_{j,t-k}$ corresponds to index $j + km$ in $\mathbf{z}_t$. Thus $[\mathbf{\Sigma}_{zy_i}]_{j+km} = (X_{j,t-k}, y_{i,t}) \neq 0$.

Now,

$$\mathbf{e}_{j,k}^\top \mathbf{A}_y \mathbf{e}_{j,k} = \sum_{\ell=1}^{L} \frac{[\mathbf{\Sigma}_{zy_\ell}]_{j+km}^2}{\sigma_{y_\ell}^2}.$$

The term for $\ell = i$ is $[\mathbf{\Sigma}_{zy_i}]_{j+km}^2 / \sigma_{y_i}^2 > 0$. Since all terms are non-negative, the sum is strictly positive.

## A.7. Proof of Proposition 5.9

By assumption, $\mathbf{s} = \mathbf{W}_s^\top \mathbf{z}$ spans $\mathcal{C} = \text{span}(\mathbf{z}_c)$ and $\mathbf{W}_s$ is injective on $\mathcal{C}$. This means there exists a function $g :^{d_s} \to \mathcal{C}$ such that $\mathbf{z}_c = g(\mathbf{s})$ (i.e., $\mathbf{z}_c$ can be recovered from $\mathbf{s}$).

Under the structural equation model $Y = f(\mathbf{z}_c) + \boldsymbol{\eta}$ with $f$ and $P(\boldsymbol{\eta})$ environment-invariant and $(\mathbf{z}_c, \boldsymbol{\eta})e$:

$$P^e(Y \mid \mathbf{s}) = P^e(Y \mid \mathbf{z}_c) \quad (\text{since } \mathbf{s} \leftrightarrow \mathbf{z}_c \text{ is bijective on } \mathcal{C}).$$

Given $\mathbf{z}_c$, the term $f(\mathbf{z}_c)$ is deterministic, so

$$Y \mid \mathbf{z}_c \overset{d}{=} f(\mathbf{z}_c) + \boldsymbol{\eta}.$$

Since $f$ does not depend on $e$, $P(\boldsymbol{\eta})$ does not depend on $e$, and $\boldsymbol{\eta}e$ (which follows from $(\mathbf{z}_c, \boldsymbol{\eta})e$), the distribution of $Y \mid \mathbf{z}_c$ is invariant across environments. Therefore $P^e(Y \mid \mathbf{s}) = P^{e'}(Y \mid \mathbf{s})$ for all $e, e' \in \mathcal{E}$.

## A.8. Proof of Proposition 5.10

The CFL algorithm consists of the following steps:

**Step 1: Time-lagged matrix construction.** For each of the $N - s$ valid time indices, we concatenate $s + 1$ vectors of dimension $m$. This requires $O((N - s) \cdot m \cdot (s + 1)) = O(N \cdot m \cdot s)$ operations.

**Step 2: Covariance estimation.** Computing $\hat{\mathbf{A}}_y$ requires: (i) sample cross-covariance $\hat{\mathbf{\Sigma}}_{zy} \in^{d \times L}$: $O(N \cdot d \cdot L)$; (ii) forming $\hat{\mathbf{A}}_y = \sum_{i=1}^{L} \hat{\mathbf{\Sigma}}_{zy_i} \hat{\sigma}_{y_i}^{-2} \hat{\mathbf{\Sigma}}_{y_iz}$: $O(L \cdot d^2)$. Computing $\hat{\mathbf{A}}_e$ requires: (i) sample means per environment: $O(N \cdot d)$; (ii) forming outer products: $O(K \cdot d^2)$. Total for covariance estimation: $O(N \cdot d^2)$ (assuming $L, K = O(d)$).

**Step 3: Generalized eigenvalue problem.** Solving $\mathbf{A}_y \mathbf{w} = \lambda \mathbf{B} \mathbf{w}$ for $d \times d$ matrices using standard algorithms (e.g., QZ algorithm) requires $O(d^3)$ operations.

**Total complexity:** $O(Nms + Nd^2 + d^3) = O(Nd^2 + d^3)$, since $d = m(s + 1)$ implies $Nms = O(Nd)$.

For $N \gg d$, the $O(Nd^2)$ covariance estimation term dominates.

# B. Additional Theoretical Results

## B.1. Optimality Conditions for CFL

**Proposition B.1** (KKT Conditions). *The generalized eigenvalue problem* (15) *is equivalent to the KKT conditions of:*

$$\max_{\mathbf{w}} \mathbf{w}^\top \mathbf{A}_y \mathbf{w} \quad s.t. \quad \mathbf{w}^\top (\mathbf{A}_e + \epsilon \mathbf{I}) \mathbf{w} = 1. \tag{24}$$

*Proof.* The Lagrangian is $\mathcal{L}(\mathbf{w}, \lambda) = \mathbf{w}^\top \mathbf{A}_y \mathbf{w} - \lambda(\mathbf{w}^\top (\mathbf{A}_e + \epsilon \mathbf{I}) \mathbf{w} - 1)$. Stationarity $\nabla_{\mathbf{w}} \mathcal{L} = 0$ gives $\mathbf{A}_y \mathbf{w} = \lambda(\mathbf{A}_e + \epsilon \mathbf{I}) \mathbf{w}$. $\square$

## B.2. Relationship to Correlation/CCA

**Proposition B.2** (Correlation Interpretation under Whitening). *When $L = 1$ and $\mathbf{A}_e = \mathbf{0}$, CFL reduces to*

$$\max_{\mathbf{w} \neq 0} \frac{(\mathbf{w}^\top \mathbf{\Sigma}_{zy})^2}{\mathbf{w}^\top \mathbf{w} \cdot \sigma_y^2}. \tag{25}$$

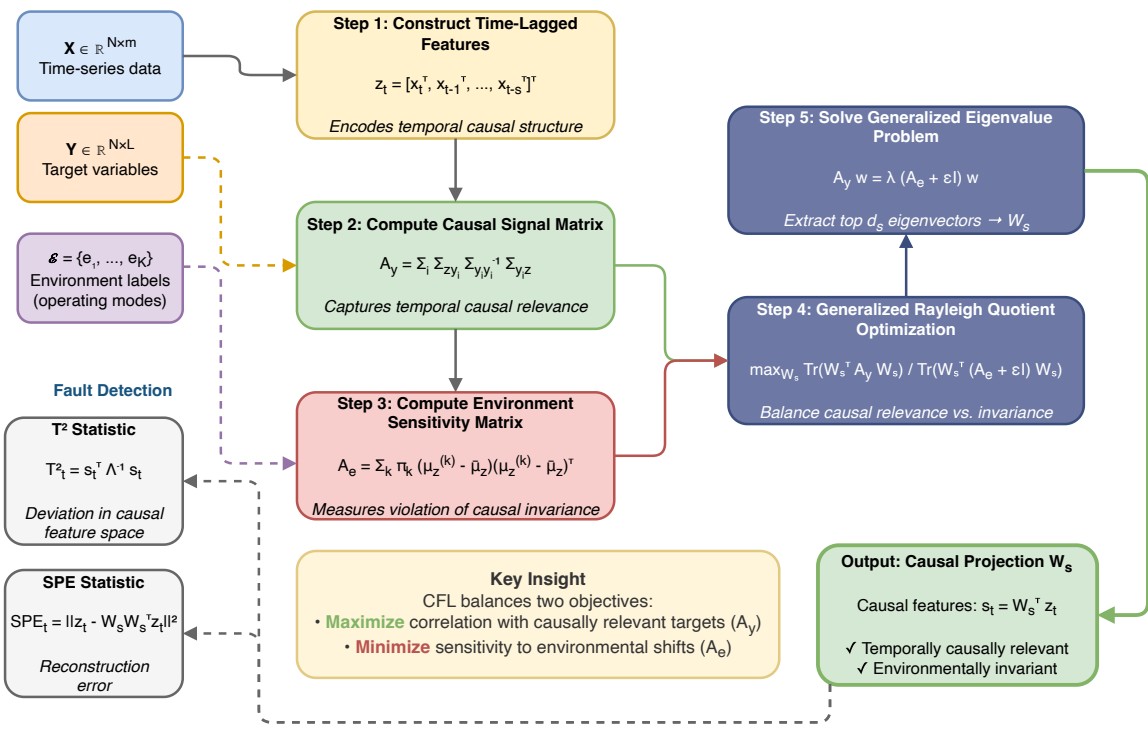

*Figure 5.* Detailed framework of causal feature learning algorithm

If additionally $(z) = I$ *(whitened $z$), this is equivalent to maximizing $Corr^2(w^\top z, y)$. Without whitening, the exact correlation objective uses $w^\top(z)w$ in the denominator.*

### B.3. Extension to Nonlinear Features

*Remark* B.3 (Kernel CFL). The CFL framework can be extended to nonlinear features via the kernel trick. Define feature map $\phi :^d \to \mathcal{H}$ and kernel $k(z, z') = \langle \phi(z), \phi(z') \rangle_{\mathcal{H}}$. Kernel CFL can penalize environment mean shifts in RKHS by matching mean embeddings, yielding kernelized analogues of $A_y$ and $A_e$.

## C. Experimental Details

Figure 5 provides a detailed illustration of the CFL algorithm pipeline. Figures 6–10 present the fault detection monitoring plots for all baseline methods. Each figure arranges all 21 faults, with method-specific monitoring statistics shown in separate rows per group. Red dashed lines indicate control limits at 95% significance level. Figure 11 summarizes the average FDR comparison across all methods.

### C.1. Tennessee Eastman Process Variables

Table 3 summarizes the Tennessee Eastman Process variables used in our experiments.

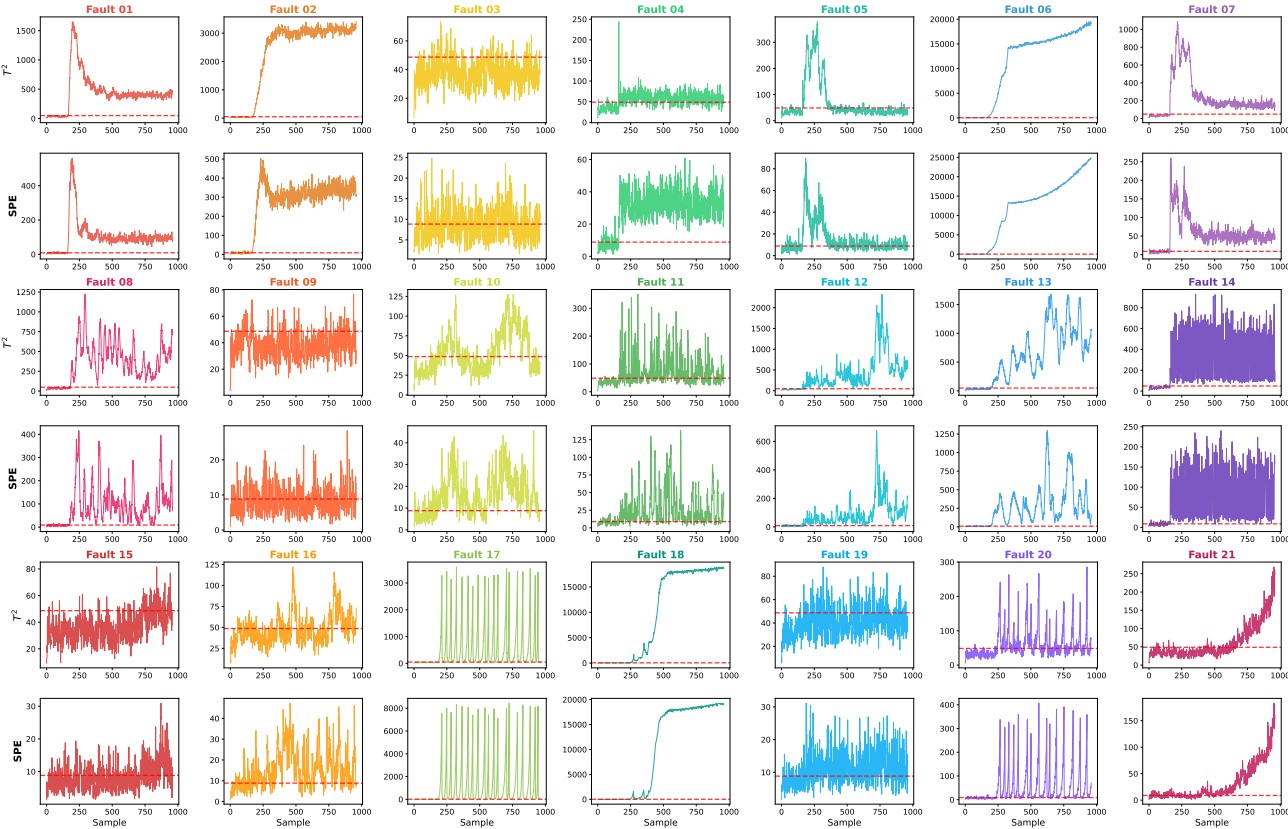

*Figure 6.* PCA fault detection monitoring results for all 21 TEP faults.

*Table 3.* Tennessee Eastman Process variables.

| Variable | Description |
| --- | --- |
| XMEAS(1-7) | Feeds and flows |
| XMEAS(8-12) | Reactor measurements |
| XMEAS(13-17) | Separator measurements |
| XMEAS(18-22) | Stripper measurements |
| XMEAS(23-41) | Compositions and analyses |
| XMV(1-11) | Manipulated variables |

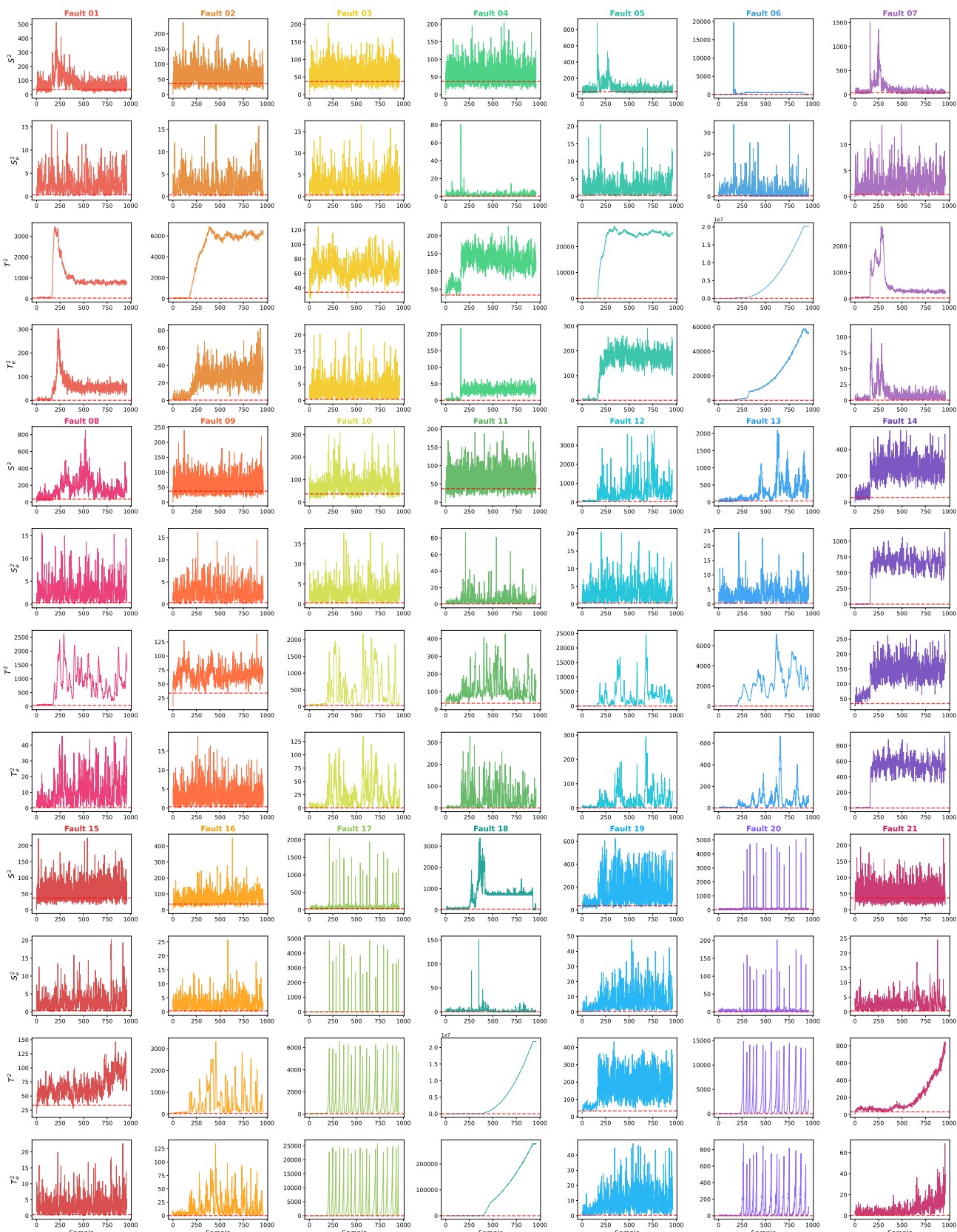

*Figure 7.* SFA fault detection monitoring results for all 21 TEP faults.

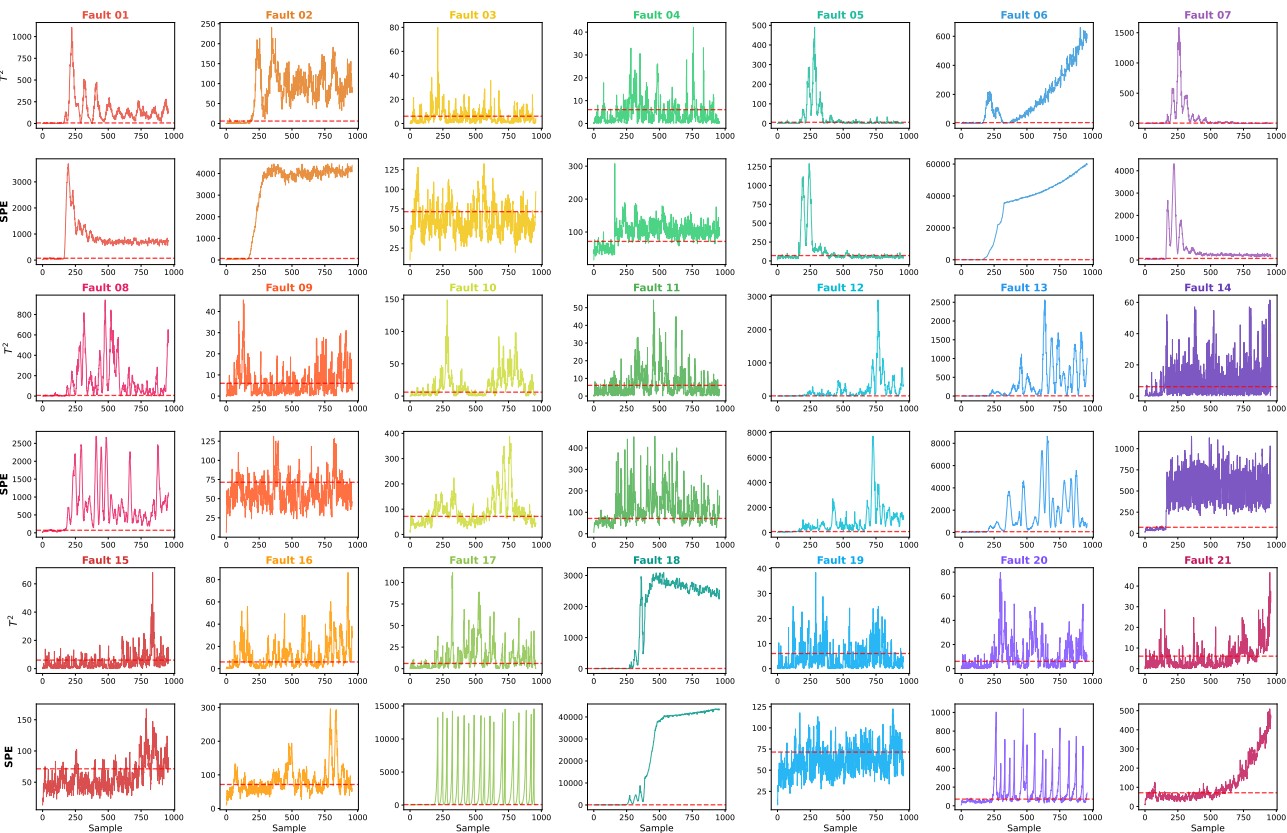

*Figure 8.* DiPLS fault detection monitoring results for all 21 TEP faults.

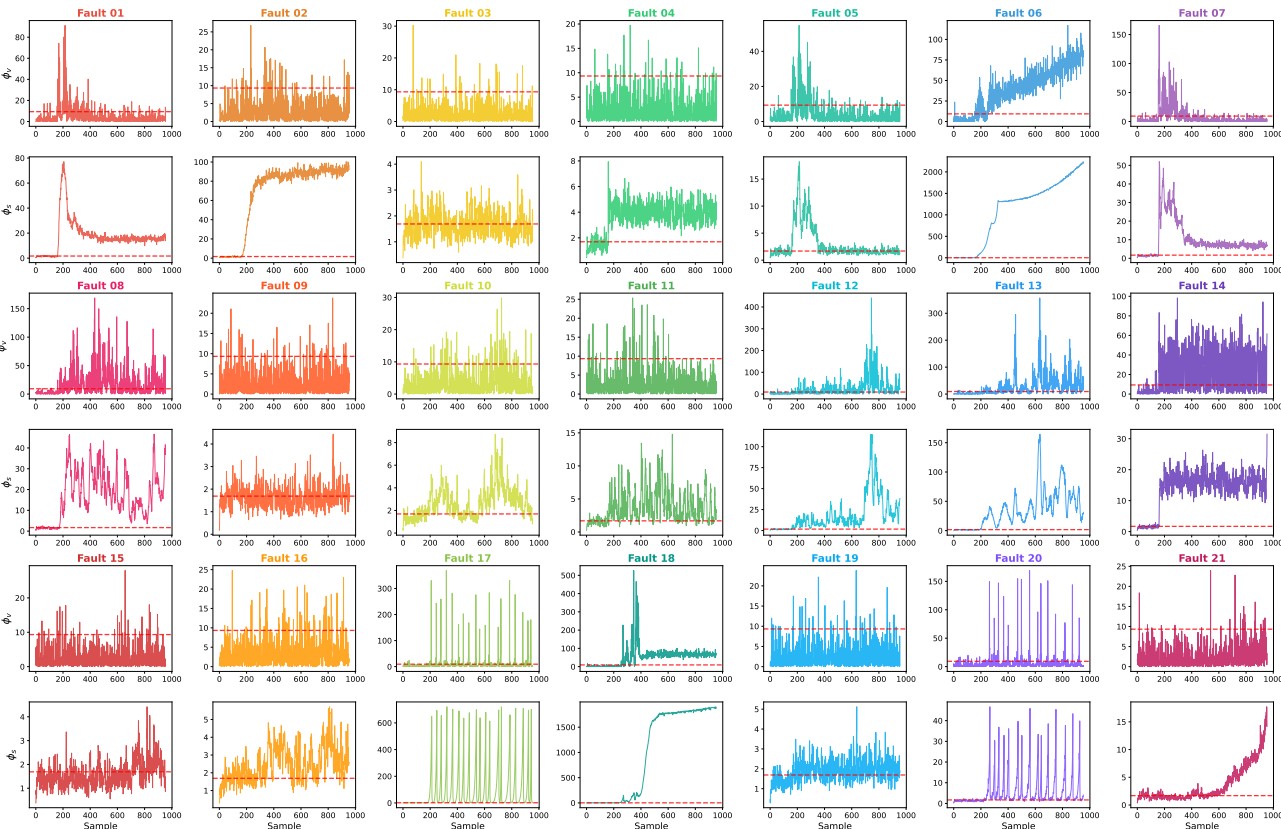

*Figure 9.* DiPCA fault detection monitoring results for all 21 TEP faults.

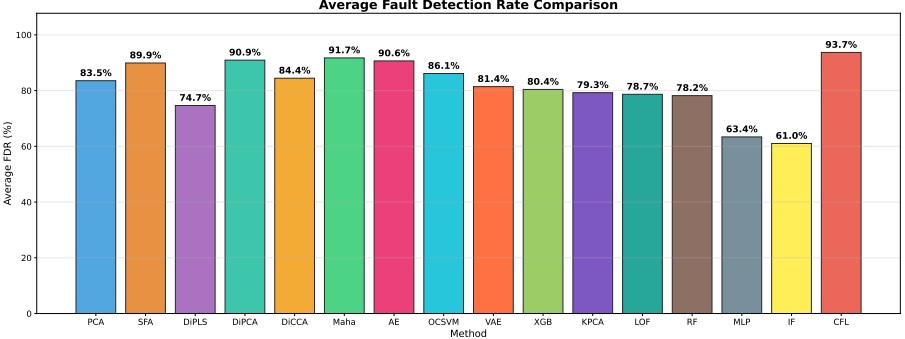

*Figure 10.* DiCCA fault detection monitoring results for all 21 TEP faults.

*Figure 11.* Average FDR comparison across all methods. CFL achieves the highest average FDR of 93.7%.

