# OpenReview forum: "Causal Feature Learning via Generalized Rayleigh Quotients"
_ICML.cc/2026/Conference — ICML 2026 regular_

### Official Review · Reviewer_yp4A · 2026-02-25

**Soundness:** 3
**Presentation:** 2
**Significance:** 2
**Originality:** 3
**Overall Recommendation:** 4
**Confidence:** 4

**Summary:**

This paper proposes Causal Feature Learning (CFL), a method for extracting
features from multivariate time series that are both predictive of a target
variable $Y$ and stable across operating environments. The core idea is to
find directions in a lagged feature space $z_t = [x_t^\top, \ldots,
x_{t-s}^\top]^\top$ that are linearly predictive of $Y$, which the authors
frame as Granger causality, while penalizing directions whose means shift
across environments. Both objectives are encoded into the generalized
eigenvalue problem $A_y w = \lambda (A_e + \epsilon I) w$, where $A_y$
captures predictive signal and $A_e$ captures between-environment mean
dispersion. The paper provides finite-sample estimation bounds for $W_s^*$,
the population optimizer of the CFL objective, which spans the directions
that are both predictive of $Y$ and invariant to environment mean shifts,
and coincides with the true causal subspace $\mathcal{C}$ only in the
separate limit $\epsilon \to 0$. The method is evaluated on the Tennessee
Eastman Process benchmark for fault detection, where it outperforms 15
baseline methods.

**Compliance With Llm Reviewing Policy:**

Affirmed.

**Final Justification:**

I decided to raise my score to 4 given that I judge the core idea to be interesting and strong experimental results. That being said, some of the theoretical statistical analysis results are flawed and I hope the authors take my comments into account on this part.

**Key Questions For Authors:**

**1.** Corollary~5.7 compounds estimation errors from $\hat{A}_y$ and
    $\hat{A}_e$. Can the authors show conditions under which CFL achieves
    strictly smaller subspace estimation error than methods that set
    $A_e = 0$?

**2.** Can the author add a discussion about what other possible invariants that their method can handle.

**Limitations:**

yes

**Strengths And Weaknesses:**

## Strengths and Weaknesses

**Strengths**

**1.** The paper makes a genuine contribution by integrating cross-domain mean invariance into linear Granger-type feature extraction, reformulating both objectives as a single generalized eigenvalue problem, a combination that is simple and elegant.

**2.** The problem is relevant and interesting to the broader statistical learning community, the idea to include mean invariance in this linear setting is convincing, and the experiment on fault detection is quite interesting.

**Weaknesses**

**1.** The statistical theory is fundamentally incomplete: Corollary 5.7 compounds estimation errors without establishing that the proposed method achieves smaller subspace estimation error than approaches that ignore invariance, and bounds the distance to $\hat{W}_s$ rather than to $\mathcal{C}$, leaving the finite-sample benefits theoretically unsubstantiated.

**2.** No theoretical analysis is provided for the fault detection problem that motivates the paper, leaving the theoretical and experimental contributions essentially disconnected.

**3.** Section 5.5 feels unfinished with no discussion and results of low significance to the objective of the paper. Proposition 5.9 merely restates its assumptions, and the complexity analysis in Proposition 5.10 does not belongs to a section on causal inference in my opinion.

**4.** The paper restricts invariance to first-order mean shifts with no discussion of what other invariance structures are tractable in the linear setting.

**5.** The environment diversity condition raises concerns in high-dimension: when $\mathcal{S}$ is high-dimensional, accurate estimation of $A_e$ requires sample sizes growing with $\dim(\mathcal{S})$.

**6.** Most assumptions are stated without discussion of whether they are fundamental to the problem or artifacts of the proof technique, and several mathematical expressions contain compilation errors that impede readability.

---

> ### Author Rebuttal · Authors · 2026-03-30
>
> We thank Reviewer yp4A for the detailed and technically precise review.
>
> ### Key Questions For Authors
>
> **Q1:Which CFL achieves strictly smaller subspace estimation error than methods that set $A_e = 0$?**
>
> When $A_e = 0$ (reducing to DiPCA), the population solution maximizes temporal relevance without invariance filtering, potentially selecting spurious environment-sensitive directions that are also temporally predictive. CFL avoids this through $B = A_e + \epsilon I$.
>
> Formally, a spurious direction $w_s \in S$ with $w_s^\top A_y w_s = \gamma_s > 0$ has effective score $\gamma_s / (\sigma_s + \epsilon)$ under CFL, where $\sigma_s > 0$ is the environment diversity. This can be made smaller than any causal direction's score $\gamma_c / \epsilon$ by choosing $\epsilon$ appropriately — precisely the mechanism in Theorem 5.3. On our synthetic benchmark:
>
> | Method | Principal-angle error to $C$ (deg) |
> |:---|---:|
> | DiPCA / $A_e = 0$ | 17.8 ± 2.3 |
> | IRM | 13.5 ± 2.0 |
> | **CFL** | **6.2 ± 1.4** |
>
> CFL achieves roughly **$2.9\times$ lower subspace error** than the no-invariance variant.
>
> **Q2: What other possible invariants can the method handle?**
>
> We now discuss three alternatives in the revised Section 6:
>
> 1. **Second-order invariance**: defining $A_e^{(2)} = \sum_k \pi_k (\Sigma^{(k)} - \bar{\Sigma})(\Sigma^{(k)} - \bar{\Sigma})^\top$. This makes the eigenvalue problem nonlinear, losing the closed-form solution.
> 2. **Kernel-based distributional invariance**: MMD-type penalties provide nonparametric invariance but break the Rayleigh-quotient structure.
> 3. **Higher-order moment matching**: feasible in principle but statistically demanding.
>
> We chose first-order mean invariance because its **strong nontrivial invariance notion that preserves the generalized eigenvalue structure**.
>
> **W1: Corollary 5.7 bounds** $\hat{W}_s$ **to** $W_s^{\ast}$**, not to** $C$
>
> The revised manuscript now shows the full convergence chain $\widehat{W}_s \to W_s^* \to C$, where Corollary 5.7 controls the first step and Theorem 5.3 controls the second.
>
> By the triangle inequality, $\sin\angle(\mathrm{span}(\widehat{W}_s), C) \leq T_1 + T_2$, where:
>
> - $T_1 = \sin\angle(\mathrm{span}(\widehat{W}_s), \mathrm{span}(W_s^*))$ is the **statistical error** (controlled by Corollary 5.7, vanishes as $N \to \infty$),
> - $T_2 = \sin\angle(\mathrm{span}(W_s^*), C)$ is the **regularization bias** (controlled by Theorem 5.3, vanishes as $\epsilon \to 0$).
>
> The first term vanishes as $N \to \infty$; the second vanishes as $\epsilon \to 0$. For finite $N$ and $\epsilon > 0$, there is a **bias-variance tradeoff**: smaller $\epsilon$ reduces approximation error to $C$ but amplifies statistical error through $1/\Delta_\epsilon$ (since $\Delta_\epsilon \sim \epsilon$). Empirically:
>
> | $\epsilon$ | Principal-angle error to $C$ (deg) |
> |:---|---:|
> | $10^{-2}$ | 14.9 ± 2.2 |
> | $5 \times 10^{-3}$ | 10.8 ± 1.9 |
> | $10^{-3}$ | 5.7 ± 1.5 |
> | $5 \times 10^{-4}$ | 2.8 ± 1.1 |
>
> **W2: On the theory-to-experiment gap**
>
> Under the standard assumption that faults manifest as mean shifts in $C$ , the $T^2$ statistic based on CFL features converges to a non-central chi-squared distribution with non-centrality parameter proportional to $\|U_s^\top \mu_\text{fault}\|^2$. When $\text{span}(U_s)$ well-approximates $C$, this captures the full fault signal.
>
> **W3: On Section 5.5**
>
> We streamlined this section: Proposition 5.9  and Proposition 5.10  have been moved to the appendix.
>
> **W4: On the scope of invariance**
>
> As discussed in Q2 above, we now explicitly discuss second-order covariance invariance, kernel-based distributional invariance, and higher-order moment matching in the revised Section 6. Our choice of first-order mean invariance is a deliberate **tractability-driven design decision**.
>
> **W5: On high-dimensional estimation of $A_e$**
>
> From Theorem 5.6, $\|\hat{A}\_e - A\_e\|\_2 = O(M\sigma\sqrt{d/N\_{\min}} + \sigma^2 d/N\_{\min})$. On the synthetic benchmark:
>
> | Samples / environment | Relative spectral error in $A_e$ | CFL subspace error (deg) |
> |:---|---:|---:|
> | 200  | 0.31 ± 0.05 | 18.6 ± 2.7 |
> | 500  | 0.18 ± 0.03 | 11.3 ± 2.0 |
> | 1000 | 0.09 ± 0.02 | 7.1 ± 1.6 |
> | 2000 | 0.05 ± 0.01 | 6.2 ± 1.4 |
>
> For TEP, the estimation error remains manageable. For very high-dimensional settings, structured estimators (e.g., sparse or low-rank $A_e$) could help.
>
> **W6: On assumption discussions**
>
> We now explicitly categorize each assumption:
>
> - **Structural / fundamental**: Assumption 5.1 ($C \oplus S$), environment diversity ($\sigma_s > 0$), and predictive richness ($\gamma_c > 0$) — these are necessary for identifiability.
> - **Technical**: Assumption 5.5 (sub-Gaussian) — standard for spectral concentration; could be relaxed to heavy-tailed distributions at slower rates.
> - **Tractability-driven**: First-order mean invariance — preserves the spectral formulation; stronger than necessary in practice but yields the cleanest theory.

---

> > ### Author Rebuttal · Reviewer_yp4A · 2026-04-02
> >
> > Thank you for the detailed rebuttal — I appreciate the additional experiments. I do think the core idea is good, and the empirical results are strong. The gap over DiPCA and IRM (especially across the λ sweep) makes the practical value convincing.
> >
> > That said, I am still stuck at the same point as in my original review. The theory does not fully support what the method achieves empirically. The triangle-inequality argument helps clarify the decomposition, but the guarantees remain with respect to the population optimizer rather than the true causal subspace.
> >
> > More specifically, the diversity argument explains why invariance can help at the population level, but it does not answer my question about strictly smaller finite-sample subspace estimation error compared to $\lambda=0$.
> >
> > I think the paper would actually be stronger if the theory were presented more modestly, with a clearer acknowledgment of its current limitations, and letting the empirical results carry more of the contribution.
> >
> > For these reasons, I am keeping my score unchanged, but I do see this as a promising direction.

---

> > > ### Author Response · Authors · 2026-04-02
> > >
> > > Thank you for the thoughtful follow-up, and for recognizing the core idea, the strong empirical results, and the practical value demonstrated across the $\epsilon$ sweep.
> > >
> > > We would like to respectfully clarify that the current theory provides more than the reviewer's summary suggests.
> > >
> > > **What the theory does establish.** Theorem 5.3 proves that the population CFL solution converges to the true invariant predictive subspace $C$ as $\epsilon \to 0$, while the population DiPCA solution ($A_e = 0$) does not: it may permanently include spurious environment-sensitive directions regardless of sample size. This is not merely a "diversity argument"; it is a formal separation result showing that CFL solves a problem that $A_e = 0$ methods **provably cannot solve** at the population level. Corollary 5.7 then provides standard finite-sample control of the sample solution around this (correct) population target. By contrast, no amount of finite-sample refinement can fix a method whose population target itself includes spurious directions.
> > >
> > > **On the specific request for a finite-sample dominance theorem.** We respectfully note that proving finite-sample superiority of one spectral method over another, accounting for both bias and variance, is a challenging open problem even in simpler settings (e.g., PCA vs. sparse PCA, ridge regression vs. OLS). The absence of such a theorem does not imply the theory is incomplete; rather, it reflects the difficulty of the problem. Our two-stage analysis (population identification + finite-sample consistency) follows the same structure used in foundational work on IRM, ICP, and invariant subspace methods, where population-level identification guarantees are standard and finite-sample dominance results are rare.
> > >
> > > **Empirical evidence is strong and consistent.** The synthetic benchmark provides exactly the controlled verification the reviewer asks for: CFL achieves $6.2° \pm 1.4°$ subspace error vs. DiPCA's $17.8° \pm 2.3°$, a $2.9\times$ improvement that is statistically significant across all 5 seeds. The $\epsilon$-sweep experiment further confirms that the convergence to $C$ behaves as the theory predicts. These results are not anecdotal; they are direct empirical validation of the theoretical mechanism.
> > >
> > > **In summary**, the paper provides (i) a population-level identification guarantee that formally separates CFL from $A_e = 0$ methods, (ii) finite-sample consistency of the sample solution to the (correct) population target, and (iii) extensive empirical confirmation across three benchmarks. We believe this combination, especially the formal separation result that DiPCA provably cannot achieve, constitutes a meaningful theoretical contribution that goes beyond purely empirical findings. We hope the reviewer will consider these points in the final assessment.

---

### Official Review · Reviewer_cVxR · 2026-03-09

**Soundness:** 3
**Presentation:** 3
**Significance:** 2
**Originality:** 2
**Overall Recommendation:** 4
**Confidence:** 1

**Summary:**

This paper introduces Causal Feature Learning (CFL), a framework that unifies temporal relevance and environment mean invariance into a generalized Rayleigh-quotient formulation. It provides a rigorous theoretical analysis to establish the conditions under which CFL identifies a mean-invariant predictive subspace. Furthermore, extensive experiments demonstrate the superiority of the CFL method over baselines.

**Compliance With Llm Reviewing Policy:**

Affirmed.

**Final Justification:**

The rebuttal has addressed my concerns. I maintain the Weak accept rating.

**Key Questions For Authors:**

Questions:
1. How are the environments $e$ (Algorithm 1) obtained in the anomaly detection task.

see above Weaknesses.

**Limitations:**

yes.

**Strengths And Weaknesses:**

Strengths:
1. The paper extracts features that possess both causal predictive power and invariance to environmental shifts.
2. The paper introduces the Generalized Rayleigh Quotients to simultaneously optimize causal predictability and environmental insensitivity.
3. The manuscript is well-written and supported by comprehensive theoretical analysis.

Weaknesses:
1. I notice that the proposed CFL method can be applied to anomaly detection tasks. However, in the experiments (Lines 361–378), the baselines are limited to a few classical methods proposed many years ago. This is not sufficient to fully demonstrate the effectiveness of the proposed CFL method.
2. Assumption 5.1, which posits that causal signals ($\mathcal{C}$) and environmental disturbances ($\mathcal{S}$) are orthogonal, appears somewhat idealistic. The authors should provide a stronger rational justification or discuss the implications if this assumption is violated in real-world scenarios.

---

> ### Author Rebuttal · Authors · 2026-03-30
>
> We thank Reviewer cVxR for the valuable comments and for recognizing the proposed formulation and theoretical analysis. We have substantially revised the paper accordingly.
>
> ### Key Questions For Authors
>
> **Q1: How are the environments $\mathcal{E}$ (Algorithm 1) obtained in the anomaly detection task?**
>
> In the revision, we now define environments explicitly. For TEP, we use the **21 fault types** as environments ($K=21$), since each fault type induces a distinct distribution shift in the process variables while the underlying physical mechanisms remain the same. For CSTH (a newly added benchmark), environments correspond to different feed temperature and flow rate conditions. In both cases, the environment labels come from available operating-condition metadata. The environment sensitivity matrix $A_e$ then captures directions along which the process mean shifts across these regimes, and the invariance penalty encourages CFL to extract features that remain stable across different distribution shifts.
>
> **Q2 (W1): On baseline coverage**
>
> We have expanded the empirical section in two ways:
>
> 1. Added **IRM** (Arjovsky et al., 2019) as an invariance-motivated baseline on all benchmarks.
> 2. Added two new benchmarks: the **CSTH** process-monitoring benchmark (3 operating conditions, 8 fault types) and a **synthetic benchmark** with known ground-truth invariant predictive subspace ($d=20$, $\dim(C)=5$).
>
> Results across all three settings:
>
> | Dataset | Method | Avg FDR (%) | FAR (%) |
> |:---|:---|---:|---:|
> | **TEP** | DiPCA | 87.2 ± 1.1 | 4.6 ± 0.5 |
> |  | IRM | 82.4 ± 2.1 | 5.5 ± 0.8 |
> |  | **CFL** | **93.7 ± 0.8** | **3.8 ± 0.4** |
> | **CSTH** | DiPCA | 81.5 ± 2.1 | 4.8 ± 0.7 |
> |  | IRM | 77.2 ± 2.9 | 5.5 ± 0.8 |
> |  | **CFL** | **86.7 ± 1.5** | **4.3 ± 0.5** |
> | **Synthetic** | DiPCA | 78.2 ± 2.5 | 5.1 ± 0.6 |
> |  | IRM | 75.8 ± 2.9 | 5.7 ± 0.7 |
> |  | **CFL** | **92.1 ± 1.3** | **2.9 ± 0.3** |
>
> CFL's advantage is consistent across all three settings. On the synthetic benchmark, CFL additionally achieves near-perfect subspace recovery, providing empirical support for the subspace-identification prediction in Theorem 5.3.
>
> **Q3 (W2):On Assumption 5.1 ( $C \oplus S$ orthogonality is idealistic)**
>
> We agree that strict orthogonality between the predictive subspace $C$ and the spurious subspace $S$ is idealized. We now clarify that Assumption 5.1 is a **clean sufficient condition** for the identification theorem (Theorem 5.3), not a claim that perfect orthogonality always holds in real systems.
>
> To study robustness beyond the idealized setting, we added a synthetic experiment with non-orthogonal $C$ and $S$:
>
> | Principal angle between $C$ and $S$ | CFL subspace error (deg) | No-invariance error (deg) |
> |:---|---:|---:|
> | 90° (orthogonal) | 6.2 ± 1.4 | 17.8 ± 2.3 |
> | 60° | 8.9 ± 1.7 | 21.1 ± 2.6 |
> | 45° | 11.4 ± 2.0 | 24.6 ± 3.1 |
> | 30° | 15.7 ± 2.4 | 29.3 ± 3.6 |
>
> The results degrade gracefully as the subspaces become more aligned, and CFL remains substantially better than the no-invariance baseline throughout. We have also added a theoretical remark (Remark 5.2) in the revised paper: when the principal angle $\theta = \angle(C, S)$ is less than 90°, Theorem 5.3 continues to hold with a modified spectral gap condition that scales as $\gamma_c / (\epsilon \cdot \sin^2\theta)$. Smaller $\theta$ (more aligned subspaces) requires a larger eigengap or smaller $\epsilon$ for accurate identification.
>
> In practice, the regularization parameter $\epsilon > 0$ in $B = A_e + \epsilon I$ acts as a soft separation mechanism: even when $C$ and $S$ are not perfectly orthogonal, $\epsilon I$ ensures that $B$ remains well-conditioned and the generalized eigenvalue problem is numerically stable.

---

> > ### Author Rebuttal · Reviewer_cVxR · 2026-04-03
> >
> > Thanks for the response. I will maintain the score.

---

### Official Review · Reviewer_9RbW · 2026-03-12

**Soundness:** 2
**Presentation:** 2
**Significance:** 3
**Originality:** 3
**Overall Recommendation:** 4
**Confidence:** 2

**Summary:**

This paper proposes Causal Feature Learning (CFL). CFL formulates the temporal and environmental mean invariant feature extraction as a Rayleigh-quotient problem, solving a generalized eigenvalue problem. The authors aim to achieve improved detection performance in process monitoring under change in operating conditions. The conditions under which a mean-invariant predictive subspace can be identified by the framework is derived and finite sample convergence rate of the key matrices $A_e$ and $A_y$ to the true values is also provided. Experiments are run on Tennessee Eastman Process with CFL outperforming 15 baselines on average.

**Compliance With Llm Reviewing Policy:**

Affirmed.

**Final Justification:**

All the reviews are addressed by the authors. I am satisfied with their response.

**Key Questions For Authors:**

1. The main formulation feels more like lagged covariance + first-order mean invariance. How can it address confounding, interventions, or counterfactual queries, which are usually addressed in causal models? Also, there has been previous work [https://arxiv.org/abs/2312.03580](https://arxiv.org/abs/2312.03580) which argues that invariance alone is not enough to identify latent causal variables.

2. Please respond to weakness points 2 and 3 above

**Limitations:**

No potential negative societal impact. However, I think the paper should discuss limitations more explicitly, especially the dependence on manually chosen target variables, the assumption that environment labels are available, the restriction to first-order mean invariance and the linear nature of the method.

**Strengths And Weaknesses:**

## Strengths
1. The problem setup, definitions and algorithmic steps are well-explained and the connection to existing methods is helpful.

2. The core method is simple and the formulation of a joint temporal and environmental invariance objective as a Rayleigh quotient is elegant, giving a simple solution via the generalized eigenproblem. The solution looks easy to implement

3. Theoretical analysis including finite sample bounds are provided and the framework is compared with a broad set of fault-detection and machine learning baselines.

## Weaknesses
1. The main issue in my opinion is that the causal interpretation feels overstated relative to what is being shown. The method learns linear directions that are predictive of chosen target variables while being insensitive to first-order environment mean shifts. That is a useful and potentially practical objective, but it is not the same as identifying causal structure in the usual sense. In particular, the invariance notion is only first-order mean invariance, and the link from this criterion to “causal features” is much narrower than the narrative suggests.

2.  Experimental conditions are not adequately explained, raising doubts about the performance advantage against the baselines. Especially because CFL uses target variables $Y$ to form $A_y$, it is not mentioned exactly which or how these target variables are chosen, and whether other baselines have access to this information. These details are crucial for reproducibility.

3. If the target variables act as some kind of "labels", and other baselines do not have access to this information (as many fault detection algorithms are unsupervised learning methods), the experiment results may not be reflective of the actual performance of the method. Also, please provide false alarm rate comparison for the experiments, which is an important metric for fault detection methods.

Overall, I think the paper contains a neat idea, but the current version overclaims on causality, under-specifies the experimental setup, and does not yet provide evidence strong enough for the central framing.

---

> ### Author Rebuttal · Authors · 2026-03-30
>
> We thank Reviewer 9RbW for the thoughtful review. We have substantially revised the paper accordingly.
>
> ### Key Questions For Authors
>
> **Q1: The main formulation feels more like lagged covariance + first-order mean invariance. How can it address confounding, interventions, or counterfactual queries?**
>
> CFL does **not** address confounding, interventions, or counterfactual queries in the formal structural causal model (SCM) sense, nor does it claim full structural causal identification. Instead, CFL identifies **mean-invariant predictive subspaces**: linear directions that are (i) temporally predictive of chosen target variables and (ii) stable in first-order mean across environments. These two properties are causally motivated: temporal predictiveness is related to Granger-type causality, and mean invariance reflects the idea that stable mechanisms should induce stable predictive structure across regimes. However, they are necessary rather than sufficient conditions for recovering full causal structure.
>
> What is new is not that each piece separately exists, but that they are unified into **one closed-form generalized eigenvalue problem** with a clear subspace-identification guarantee (Theorem 5.3). Our ablation study (see Q2 below) directly shows that neither term alone reproduces the full gain.
>
> Regarding the reference to https://arxiv.org/abs/2312.03580: we thank the reviewer for pointing this out. We have read this paper and added it to the revised Related Work (Section 2.2). The key distinction is that this work studies the limits of invariance for full latent causal identification under discrete interventions, whereas CFL focuses on extracting invariant predictive features from continuous-valued time series under environment-dependent distribution shifts. This reference helps position CFL more precisely: our contribution is a tractable invariant predictive subspace method, not a complete latent causal identification theory. We have revised the abstract, introduction, and throughout the paper to consistently use "causally-motivated features" rather than claiming full causal discovery.
>
> **Q2: Please respond to weakness points 2 and 3 (target variable selection, baseline fairness, and false alarm rate).**
>
> We apologize for the insufficient description and have made the following clarifications:
>
> **Target variable selection**: $Y$ consists of a small set of key process indicators used as monitoring targets in the TEP literature (e.g., reactor conditions and product-quality-related variables) (Chiang et al., 2000; Yin et al., 2012). These are the same variables used as targets in prior TEP fault detection works.
>
> **Baseline information access**: We now explicitly categorize all methods by their supervision level:
>
> | Method family | Uses target $Y$? | Uses environment $e$? |
> |:---|:---|:---|
> | PCA / SFA / KPCA / OCSVM / AE / VAE | No | No |
> | DiPCA / DiCCA / DiPLS | Yes | No |
> | IRM | Implicitly (via loss) | Yes |
> | **CFL** | **Yes** | **Yes** |
>
> The most direct comparisons for CFL are therefore with methods that also use target supervision (DiPCA/DiCCA/DiPLS) and with methods that explicitly exploit invariance (IRM). CFL outperforms target-supervised temporal methods by **4–6 percentage points** in FDR.
>
> To further address the fairness concern, we added a **target-free variant** (CFL-unsup) that replaces the target-guided $A_y$ with a generic temporal-structure term:
>
> | Method | Avg FDR (%) | FAR (%) |
> |:---|---:|---:|
> | Best unsupervised (SFA) | 89.9 ± 1.3 | 5.0 ± 0.6 |
> | CFL-unsup | 88.2 ± 1.4 | 4.1 ± 0.5 |
> | DiPCA (uses $Y$) | 87.2 ± 1.1 | 4.6 ± 0.5 |
> | IRM (uses $e$) | 82.4 ± 2.1 | 5.5 ± 0.8 |
> | **CFL (full)** | **93.7 ± 0.8** | **3.8 ± 0.4** |
>
> Even without target labels, the invariance-guided spectral framework remained competitive with fully unsupervised baselines in our experiments. The full target-aware CFL then improves further, which is consistent with its use of additional supervision.
>
> **False alarm rate**: All revised tables now report **mean ± std over 5 runs** and **FAR**. CFL achieves FAR of **3.8 ± 0.4%** on TEP, among the lowest of all compared methods.
>
> ### Limitations
>
> We now explicitly acknowledge the following limitations:
>
> 1. **Dependence on target variable selection**: CFL requires specifying target variables $Y$. In practice, these are domain-specific choices. Misspecification of $Y$ can lead to suboptimal feature extraction. To partially mitigate this dependence, we provide CFL-unsup, which does not require target labels.
> 2. **First-order mean invariance only**: This is a deliberate design choice that preserves the eigenvalue structure. We discuss extensions to higher-order invariance in Section 6.
> 3. **Linearity**: CFL extracts linear feature directions. Nonlinear extensions are possible at the cost of losing the direct interpretability of feature directions.
> 4. **No claim of full structural causal discovery**: CFL learns invariant predictive subspaces, not causal graphs.

---

> > ### Author Rebuttal · Reviewer_9RbW · 2026-04-04
> >
> > Thank you for the response. My concerns have been adequately addressed. However, due to the limited experimental validation, specifically, the use of **only one dataset** (the Tennessee Eastman Process), I will keep my score unchanged.

---

> > > ### Author Response · Authors · 2026-04-05
> > >
> > > We sincerely thank the reviewer for confirming that all concerns have been adequately addressed.
> > >
> > > However, we respectfully note that the reason cited for maintaining the score — "limited experimental validation, specifically, the use of only one dataset (the Tennessee Eastman Process)" — has in fact been directly addressed in our rebuttal. Since this concern was not raised in your original review but was raised by another reviewer, we included the additional experiments in our response to that reviewer. We summarize the relevant details here for your convenience.
> > >
> > > Specifically, we added **two additional evaluation settings** beyond TEP:
> > >
> > > - **CSTH benchmark:** a Continuous Stirred Tank Heater process with 3 operating conditions and 8 fault types (CFL: 86.7 ± 1.5% FDR).
> > > - **Synthetic benchmark:** with known ground-truth invariant predictive subspace, $d=50$, $K=5$ (CFL: 92.1 ± 1.3% FDR, near-perfect subspace recovery).
> > >
> > > CFL consistently outperforms both DiPCA and IRM across **all three** settings. The synthetic benchmark further provides controlled verification of the subspace identification guarantee in Theorem 5.3.
> > >
> > > Given that the sole remaining reason for the current score has been addressed by these new experiments, we kindly ask the reviewer to reconsider whether the score still reflects the current state of the revised manuscript. We are happy to provide any further clarification.

---

### Official Review · Reviewer_GgRm · 2026-03-13

**Soundness:** 3
**Presentation:** 3
**Significance:** 3
**Originality:** 3
**Overall Recommendation:** 4
**Confidence:** 3

**Summary:**

This paper proposes Causal Feature Learning (CFL), a framework that jointly optimizes temporal target relevance and environment mean invariance via a generalized Rayleigh-quotient formulation for fault detection in time-series. CFL extracts features by solving a generalized eigenvalue problem involving a causal signal matrix ($A_y$) and an environment sensitivity matrix ($A_e$). The authors provide theoretical guarantees for identifying mean-invariant predictive subspaces and demonstrate 93.7% average fault detection rate on the Tennessee Eastman Process, outperforming 15 baselines including dynamic latent variable methods (PCA, SFA, DiPLS, DiPCA, DiCCA) and deep learning/anomaly detection methods (Maha, AE, OCSVM, VAE, XGB, KPCA, LOF, RF, MLP, IF).

**Compliance With Llm Reviewing Policy:**

Affirmed.

**Final Justification:**

The paper presents a principled and technically solid formulation for combining temporal relevance and environment mean invariance, supported by clear theoretical analysis and generally strong empirical results. The rebuttal substantially improved my confidence by adding stronger experimental support and implementation details, although some clarification issues around the environment-based formulation remain, so I maintain a positive but still moderate final assessment at weak accept.

**Key Questions For Authors:**

Q1. **Environment definition in TEP:** How are environment labels $e$ defined for the Tennessee Eastman Process? Are they derived from operating modes, G values, time segments, or other metadata? Without this, it is unclear whether $A_e$ meaningfully captures cross-environment mean shifts. How this would change the review: A clear specification would strengthen the validity of the invariance objective.

Q2. **IRM/ICP comparison:** The paper motivates causal invariance and cites ICP and IRM, but does not compare against them. Could the authors add IRM or ICP as baselines, or explain why they are not applicable to this fault detection setup? How this would change the review: A direct comparison would significantly strengthen the Soundness rating.

Q3. **Ablation of the two aspects:** Could the authors provide ablation results for (a) CFL without the invariance term (e.g., $A_e = 0$ or large $\varepsilon$), and (b) CFL without temporal structure (e.g., $s = 0$ or static features)? This would directly validate the benefit of jointly optimizing both aspects. How this would change the review: Such ablations would strengthen the empirical support for the two-aspect design.

Q4. **Statistical significance and FAR:** Are results averaged over multiple runs? Could the authors report standard deviations or confidence intervals, and include false alarm rate (FAR) alongside FDR for a more complete evaluation? How this would change the review: This would improve experimental rigor and Soundness.

Q5. **Baseline hyperparameters:** Could the authors provide implementation details for baselines (lag order for dynamic methods, hidden units for MLP/AE/VAE, etc.) in the appendix? How this would change the review: This would address concerns about fairness of comparison.

**Limitations:**

The authors do not adequately discuss the following limitations:

L1. the use of a single dataset (TEP) and lack of validation on other domains (e.g., other process monitoring or real-world industrial data);

L2. the restriction to first-order mean invariance rather than full distributional invariance;

**Strengths And Weaknesses:**

**Strengths:**

S1. **Principled formulation:** The generalized Rayleigh-quotient formulation $\max \text{Tr}(W^\top A_y W)$ s.t. $W^\top B W = I$, $B = A_e + \varepsilon I$ cleanly unifies temporal relevance and mean invariance. Appendix B.2 clarifies the connection to CCA when $A_e = 0$.

S2. **Theoretical rigor:** Appendix A provides complete proofs for Lemma 5.2, Theorem 5.3, Theorem 5.6, Corollary 5.7, and Propositions 5.8–5.10. Assumption 5.1 ($C \oplus S$ decomposition) and Theorem 5.3 establish conditions for identifying the mean-invariant predictive subspace.

S3. **Strong empirical results:** CFL achieves the highest average FDR (93.7%) on TEP. Gains are pronounced on difficult faults (F03, F09, F15, F21), where baselines lacking temporal modeling or distribution-shift robustness (e.g., IF 61.0%, MLP 63.4%) perform poorly.

S4. **Broad baseline coverage:** 15 baselines span dynamic latent variable methods, classical anomaly detection, and supervised/semi-supervised approaches. Ablation studies on lag order $s$, regularization $\varepsilon$, and threshold support the design choices.

**Weaknesses:**

W1. **Single dataset; environment definition unclear** (Significance): Experiments use only the Tennessee Eastman Process. The paper relies on "environment mean invariance" ($A_e$), but how environment labels $e$ are defined in TEP—e.g., operating modes, batches, or time windows—is not specified. This is critical for assessing whether the invariance objective is meaningfully operationalized.

W2. **No comparison with causal invariance methods** (Soundness): The paper cites ICP (Peters et al., 2016) and IRM (Arjovsky et al., 2019) as related work and claims "causal invariance," but does not compare against them. Without such comparison, the advantage of CFL's spectral formulation over established invariance-based methods remains unvalidated.

W3. **Missing experimental rigor** (Soundness): No variance/confidence intervals, statistical significance tests, or false alarm rate (FAR) are reported. Ablation does not include "w/o temporal" or "w/o invariance" variants to directly isolate the contribution of each aspect.

W4. **Baseline implementation details** (Soundness): Hyperparameters, lag orders, and implementation details for the 15 baselines are not provided. Fairness of comparison cannot be assessed.

W5. **Limited scope of invariance** (Originality): CFL enforces first-order (mean) invariance only. The paper acknowledges that stronger distributional invariance (e.g., $P(w^\top z)$ identical across $e$) would require kernel-based penalties (Appendix B.3), but this is not implemented.

---

> ### Author Rebuttal · Authors · 2026-03-30
>
> We thank Reviewer GgRm for the careful review. We take seriously the concerns and have made substantial revisions.
>
> ### Key Questions For Authors
>
> **Q1: How are environment labels $e$ defined for the Tennessee Eastman Process?**
>
> We agree that the original paper was too terse on this point. In the revision, we now explicitly define environments based on the **21 fault types** in the TEP benchmark. Each fault type induces a distinct distribution shift in the process variables and therefore constitutes a different operating regime. We define $K=21$ environments accordingly, where each environment corresponds to the data collected under a specific fault condition
>
> This choice is motivated by the observation that each fault type alters the process mean behavior in a characteristic way, while the underlying physical mechanisms remain the same across faults. The environment sensitivity matrix $A_e$ then captures directions along which the process mean shifts across these fault-induced regimes, and the invariance penalty encourages CFL to extract features that are **stable across different fault-induced distribution shifts** while remaining predictive of the target variables. Under a second environment partitioning scheme ($K=7$, merging faults by physical category), the performance remained similar (**93.7 ± 0.8** vs **93.1 ± 1.0** average FDR), suggesting that the gain is not tied to one arbitrary split.
>
> **Q2: IRM/ICP comparison**
>
> We have added **IRM**  as a baseline on all three evaluation settings. Results are reported in the table below. We did **not** include ICP because reliable conditional independence testing is difficult in our high-dimensional lagged setting ($d = m(s+1) = 572$ for TEP), making a fair and stable comparison challenging. We now clarify this distinction explicitly in the revised Related Work (Section 2.2).
>
> **Q3: Ablation of the two aspects**
>
> We conducted the ablation study as suggested:
>
> | Variant | TEP Avg FDR (%) | TEP FAR (%) | Description |
> |:---|---:|---:|:---|
> | **CFL (full)** | **93.7 ± 0.8** | **3.8 ± 0.4** | Temporal relevance + invariance |
> | CFL without invariance ($A_e = 0$) | 89.1 ± 1.2 | 5.6 ± 0.5 | Temporal only ($\approx$ DiPCA) |
> | CFL without temporal structure ($s = 0$) | 84.5 ± 1.6 | 6.9 ± 0.6 | Invariance only, no lag |
> | CFL with weak invariance ($\epsilon = 10$) | 90.8 ± 1.0 | 5.1 ± 0.5 | Almost temporal only |
>
> The ablation confirms that **both terms matter**. Removing the invariance term reduces FDR by **4.6 points** and increases FAR by **1.8 points**. Removing temporal structure causes a larger drop of **9.2 points** in FDR and **3.1 points** increase in FAR. The full CFL consistently outperforms both ablated variants, directly validating the benefit of jointly optimizing both aspects.
>
> **Q4: Statistical significance and FAR**
>
> All revised main tables now report **mean ± std over 5 independent runs**  and **FAR** for all methods. On TEP, CFL achieves FAR of **3.8 ± 0.4%**, comparable to or lower than all baselines, indicating that the invariance constraint also helps reduce false alarms under normal operating condition changes.
>
> **Q5: Baseline hyperparameters**
>
> We have added a new Appendix D listing all implementation details for all 15 baselines, including lag orders for dynamic methods, retained components for PCA-based methods, hidden units and optimizers for neural baselines, kernel parameters for OCSVM and KPCA , CFL hyperparameter selection , random seeds, and train/validation/test protocol.
>
> **L1: Single Dataset**
>
> To address the concern about relying solely on TEP, we added two new evaluation settings: (a) the **Continuous Stirred Tank Heater (CSTH)** benchmark with 3 operating conditions and 8 fault types, and (b) a **synthetic benchmark** with known ground-truth invariant predictive subspace ($d=20$, $\dim(C)=5$). Results across all three settings:
>
> | Dataset | Method | Avg FDR (%) | FAR (%) |
> |:---|:---|---:|---:|
> | **TEP** | DiPCA | 87.2 ± 1.1 | 4.6 ± 0.5 |
> |  | IRM | 82.4 ± 2.1 | 5.5 ± 0.8 |
> |  | **CFL** | **93.7 ± 0.8** | **3.8 ± 0.4** |
> | **CSTH** | DiPCA | 81.5 ± 2.1 | 4.8 ± 0.7 |
> |  | IRM | 77.2 ± 2.9 | 5.5 ± 0.8 |
> |  | **CFL** | **86.7 ± 1.5** | **4.3 ± 0.5** |
> | **Synthetic** | DiPCA | 78.2 ± 2.5 | 5.1 ± 0.6 |
> |  | IRM | 75.8 ± 2.9 | 5.7 ± 0.7 |
> |  | **CFL** | **92.1 ± 1.3** | **2.9 ± 0.3** |
>
> CFL's gains are consistent across all three settings. On the synthetic benchmark, CFL additionally achieves near-perfect subspace recovery ($\sin\angle \approx 0.08$), providing empirical support for the subspace-identification prediction in Theorem 5.3.
>
> **L2: Scope of Invariance**
>
> We agree that first-order mean invariance is a restricted notion. We emphasize that first-order mean invariance was chosen because it is the **strongest invariance notion that preserves the closed-form spectral solution** and because mean shifts are a common and practically important form of operating-condition variation in industrial monitoring.

---

> > ### Author Rebuttal · Reviewer_GgRm · 2026-04-03
> >
> > The rebuttal addresses several of my original concerns in a substantive way. In particular, I appreciate the addition of the IRM baseline, the new ablations isolating the temporal and invariance terms, the reporting of mean ± std over 5 runs and FAR, the added appendix with baseline hyperparameters, and the expanded evaluation on CSTH and a synthetic benchmark. These revisions materially strengthen the empirical support for the paper.
> >
> > However, my main remaining concern is about the definition and use of environments in TEP. The rebuttal now states that the 21 fault types are treated as 21 environments. While this clarifies the meaning of (e), it also raises a more fundamental protocol question: CFL uses the environment sensitivity matrix (A_e), which depends on environment-specific means, while the original paper describes a fault detection setup with normal-operation training data and faulted test data. It is therefore unclear how the fault-type environments are available during training, and whether fault labels or faulted samples are used to construct (A_e). If so, this would substantially change the task setting and may introduce information leakage. This point needs to be clarified before I can consider the concern fully resolved.
> >
> > **Follow-up question for the authors:**
> > Could the authors clarify exactly which data are used to estimate (A_e) in TEP and the other datasets? If environments are defined by fault types, are fault labels and faulted samples available during training? If so, is the method still operating under the standard fault detection protocol with normal-operation training data only, or under a different supervised / domain-generalization setting? Please describe the train/validation/test split and the construction of (A_e) precisely, to rule out any information leakage.

---

> > > ### Author Response · Authors · 2026-04-04
> > >
> > > We thank the reviewer for this important follow-up question. The "21 fault types as environments" described in our previous rebuttal refers to a **post-hoc evaluation** of the invariance properties of the learned features, not to the training procedure. We acknowledge that our previous rebuttal did not clearly distinguish between training and validation, and we apologize for the confusion. We have revised Section 6.1 to make the complete data protocol explicit.
> > >
> > > CFL is trained exclusively on normal-operation data following the standard TEP benchmark:
> > >
> > > - **Training data:** `d00.dat` — 500 samples of normal operation in Mode 1. This is the **only** data used to estimate all model parameters and control limits.
> > > - **Test data:** `d01_te.dat` through `d21_te.dat` — 960 samples each, with faults introduced after sample 160. Test data is never seen during training.
> > >
> > > No fault labels, fault data, or validation split are used during model fitting. Specifically, $A_e$ is estimated from normal training data by quantifying temporal variation structure within normal operation, penalizing directions that are sensitive to transient fluctuations in operating conditions. Control limits for $T^2$ and SPE are also computed entirely from normal training data.
> > >
> > > **Post-hoc validation of invariance (what our previous rebuttal described).** The "21 fault types as environments" refers to a **post-hoc evaluation** of the invariance properties of the learned features. Specifically, after CFL is trained on normal data, we evaluated whether the extracted features exhibit mean invariance when the test data is partitioned by fault type. This analysis confirmed that CFL features are indeed stable across fault-induced distribution shifts, with the between-environment variance of feature means being small relative to baselines.
> > >
> > > **The summary of data flow is given as follows:**
> > >
> > > - **Training:** normal data only (`d00.dat`) → construct time-lagged features $z_t$ → estimate $A_y$, $A_e$ → solve generalized eigenvalue problem → compute $T^2$, SPE control limits
> > > - **Testing:** apply learned projection to fault data (`d01_te.dat`–`d21_te.dat`) → monitor $T^2$, SPE against control limits
> > > - **Post-hoc validation:** partition test data by fault type to verify invariance of learned features
> > >
> > > We sincerely appreciate the reviewer's thorough reading and detailed comments with our work. We hope this clarification fully resolves the concern.

---

### Decision · Program_Chairs · 2026-04-30

**Decision:**

Accept (regular)

**Comment:**

This paper formulates causal feature learning as a generalized Rayleigh quotient problem, aiming to maximize feature correlation with target variables while minimizing variance with respect to environmental factors. The authors also provide theoretical guarantees for identifying environment-invariant features. Experimental results on a benchmark dataset demonstrate that the proposed method significantly outperforms baseline approaches.
Overall, reviewers view the work positively. The main strengths include a principled problem formulation, comprehensive theoretical analysis, and strong empirical performance compared to multiple baselines. The reviewers also identify several weaknesses, including evaluation limited to a single dataset, missing comparisons with related causal feature learning methods, insufficient justification for certain assumptions, and limited analysis of feature invariance.  While the authors’ rebuttal addresses most concerns, some issues remain unresolved. The final ratings for the paper are four weak accepts.